**RESEARCH**

# Single-cell proteomic and transcriptomic analysis of macrophage heterogeneity using SCoPE2

Harrison Specht[1*], Edward Emmott[1,2], Aleksandra A. Petelski[1], R. Gray Huffman[1], David H. Perlman[1,3], Marco Serra[4], Peter Kharchenko[4], Antonius Koller[1] and Nikolai Slavov[1*]

* Correspondence: hms89@cornell.edu; nslavov@alum.mit.edu; nslavov@northeastern.edu
[1]Department of Bioengineering and Barnett Institute, Northeastern University, Boston, MA 02115, USA
Full list of author information is available at the end of the article

## Abstract

**Background:** Macrophages are innate immune cells with diverse functional and molecular phenotypes. This diversity is largely unexplored at the level of single-cell proteomes because of the limitations of quantitative single-cell protein analysis.

**Results:** To overcome this limitation, we develop SCoPE2, which substantially increases quantitative accuracy and throughput while lowering cost and hands-on time by introducing automated and miniaturized sample preparation. These advances enable us to analyze the emergence of cellular heterogeneity as homogeneous monocytes differentiate into macrophage-like cells in the absence of polarizing cytokines. SCoPE2 quantifies over 3042 proteins in 1490 single monocytes and macrophages in 10 days of instrument time, and the quantified proteins allow us to discern single cells by cell type. Furthermore, the data uncover a continuous gradient of proteome states for the macrophages, suggesting that macrophage heterogeneity may emerge in the absence of polarizing cytokines. Parallel measurements of transcripts by 10× Genomics suggest that our measurements sample 20-fold more protein copies than RNA copies per gene, and thus, SCoPE2 supports quantification with improved count statistics. This allowed exploring regulatory interactions, such as interactions between the tumor suppressor p53, its transcript, and the transcripts of genes regulated by p53.

**Conclusions:** Even in a homogeneous environment, macrophage proteomes are heterogeneous. This heterogeneity correlates to the inflammatory axis of classically and alternatively activated macrophages. Our methodology lays the foundation for automated and quantitative single-cell analysis of proteins by mass spectrometry and demonstrates the potential for inferring transcriptional and post-transcriptional regulation from variability across single cells.

## Introduction

Tissues and organs are composed of functionally specialized cells. This specialization of single cells often arises from the protein networks mediating physiological functions. Yet, our ability to comprehensively quantify the proteins comprising these networks in single cells has remained relatively limited [1, 2]. As a result, the protein levels in single

cells are often inferred from indirect surrogates—sequence reads from their corresponding mRNAs [3, 4].

Single-cell RNA sequencing methods have illuminated cellular types and states comprising complex biological tissues, aided the discovery of new cell types, and empowered the analysis of spatial organization [3, 5]. These methods depend on the ability to capture and detect a representative set of cellular transcripts. Many transcripts are present at low copy numbers, and with the existing scRNA-seq protocols capturing around 10–20% of molecules in a cell, the resulting sampling is very sparse for many transcripts. Because of this, the estimates of mRNA abundances are notably affected by sampling (counting) errors [3, 4].

Sampling many protein copies per gene may be feasible since most proteins are present at over 1000-fold more copies per cell than their corresponding transcripts [1, 6, 7]. This high abundance also obviates the need for amplification. Since amplification may introduce noise, obviating amplification is a desirable aspect. Thus, the high copy number of proteins may allow their quantification without amplification.

However, most technologies for quantifying proteins in single cells rely on antibodies, which afford only limited specificity [1, 8]. Although tandem mass spectrometry (MS/MS) combined with liquid chromatography (LC) and electrospray ionization (ESI) has enabled accurate, high specificity, and high-throughput quantification of proteins from bulk samples [9–12], its application to single cells is in its infancy [2, 8].

To apply these powerful MS technologies to the analysis of single cells, we developed Single-Cell ProtEomics by Mass Spectrometry (SCoPE-MS) [6, 8, 13]. SCoPE-MS introduced the concept of using isobaric carrier, which serves three important roles: (i) reducing sample loss, (ii) enhancing the detectability of ions during MS1 survey scans, and (iii) providing fragment ions for peptide sequence identification. By combining this concept with MS-compatible cell lysis, we established the feasibility of applying multiplexed LC-ESI-MS/MS to quantify proteins from single cells. Parallel efforts have focused on the label-free analysis of individual cells that does not require isobaric labeling but analyzes fewer cells per unit time [14–17].

While SCoPE-MS and its ideas have been reproduced and adopted by others [17–23], the cost, throughput, and reliability of the data fall short of our vision of single-cell proteomics [6, 8]. Our vision requires quantifying thousands of proteins and proteoforms across thousands of single cells at an affordable cost and within a reasonable time frame. Such data could support clinical applications such as biomarker discovery. Moreover, these data could permit inferring direct causal mechanisms underlying the functions of protein networks [8]. The more cells and proteoforms are quantified, the fewer assumptions are needed for this analysis. Thus, our goal in developing SCoPE2 was to increase the number of cells and proteins analyzed at an affordable cost while sampling a sufficient number of ion copies per protein to make quantitative measurements.

To achieve this goal, we followed previously outlined opportunities [6]. In particular, we over-hauled multiple experimental steps, including cell isolation, lysis, and sample preparation [2, 24]. Furthermore, we developed methods for optimizing the acquisition of MS data (DO-MS; Data-driven Optimization of MS) [25] and for interpreting these data once acquired, e.g., for enhancing peptide identification (DART-ID; Data-driven Alignment of Retention Times for IDentification) [26]. These advances combine

synergistically into a next-generation SCoPE-MS version, SCoPE2, that affords substantially improved quantification and throughput.

SCoPE2 enabled us to ask fundamental questions: do homogeneous monocytes produce homogeneous macrophages in the absence of polarizing cytokines? Are macrophages inherently prone to be heterogeneous, or is their heterogeneity simply reflecting different progenitors and polarizations induced by different cytokines? These questions are cornerstones to our understanding of macrophage heterogeneity that plays important roles in human pathophysiology: depending on their polarization, macrophages can play pro-inflammatory (usually ascribed to M1 polarization) or anti-inflammatory roles (usually ascribed to M2 polarization) and be involved in tissue development and maintenance [27]. Some studies suggest that rather than separating into discrete functional classes, the M1 and M2 states represent the extremes of a wider spectrum of end-states [27, 28]. We found that the individual macrophage-like cells were heterogeneous even though they originated from homogeneous monocytes exposed to identical environmental conditions. This heterogeneity correlates to a polarization axis that was previously described to emerge in the presence of cytokines.

## Results

The overall work-flow of SCoPE2 is illustrated in Fig. 1a. Single cells are isolated in individual wells, lysed, and their proteins digested to peptides. The peptides from each single cell are covalently labeled (barcoded) with isobaric tandem-mass-tags (TMT), and therefore, labeled peptides with the same sequence (and thus mass) appear as a single mass/charge cluster in the MS1 scans. The MS instrument isolates such clusters and fragments them, see Fig. 2. In addition to generating peptide fragments which facilitate peptide identification, fragmentation generates reporter ions (RI), whose abundances reflect protein abundances in the corresponding samples (single cells) (Fig. 1a). Below, we summarize the key advances of SCoPE2 over SCoPE-MS.

### Automated and miniaturized cell lysis

Instead of lysing cells by focused acoustic sonication, SCoPE2 lyses cells by Minimal ProteOmic sample Preparation (mPOP) [24]. mPOP uses a freeze-heat cycle that extracts proteins efficiently in pure water, thus obviating cleanup before MS analysis. mPOP allows sample preparation in multiwell plates, which enables simultaneous processing of many samples in parallel with inexpensive PCR thermocyclers and liquid dispensers. This advance over SCoPE-MS allows SCoPE2 to decrease lysis volumes 10-fold, from 10 to 1 μl, to reduce the cost of consumables and equipment over 100-fold, and to increase the throughput of sample preparation over 100-fold by parallel processing.

### Improved data integration across single cells

SCoPE2 also introduces a reference channel composed of a reference sample used in all sets. While commonly used in bulk proteomic experiments to integrate quantitative proteomic measurements across multiple multiplexed experiments, this concept was not implemented in SCoPE-MS. In SCoPE2, the reference is about 5-fold more abundant than a single-cell proteome so that the higher abundance results in improved ion-

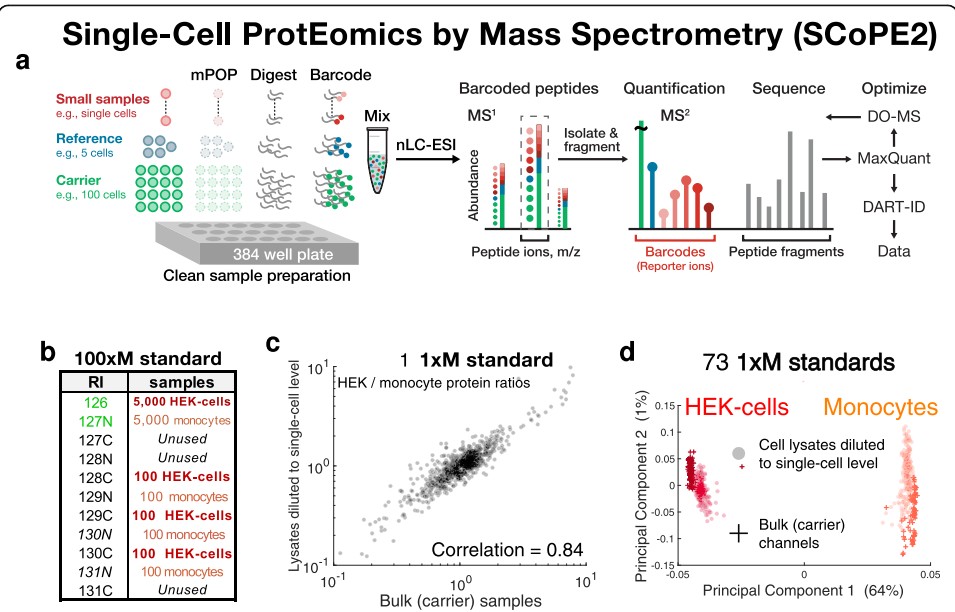

**Fig. 1** Optimizing and benchmarking MS analysis with bulk standards modeling SCoPE2 sets. **a** Conceptual diagram and work flow of SCoPE2. Cells are sorted into multiwell plates and lysed by mPOP [24]. The proteins in the lysates are digested with trypsin; the resulting peptides labeled with TMT, combined, and analyzed by LC-MS/MS. SCoPE2 sets contain reference channels that allow merging single cells from different SCoPE2 sets into a single dataset. The LC-MS/MS analysis is optimized by DO-MS [25], and peptide identification enhanced by DART-ID [26]. **b** Schematic for the design of a 100xM bulk standards. Monocytes (U937 cells) and embryonic kidney cells (HEK-293) were serially diluted to the indicated cell numbers, lysed, digested, and labeled with tandem-mass tags having the indicated reporter ions (RI). **c** Comparison of protein fold change between the embryonic kidney cells and monocytes estimated from the small samples and from the carrier samples of a 1xM standard, i.e., 1% sample from the 100xM standard described in **a**. The relative protein levels measured from bulk samples diluted to single-cell levels are very similar to the corresponding estimates from the isobaric carrier (bulk) samples. **d** Principal component analysis separates samples corresponding to embryonic kidney cells (HEK-293) or to monocytes (U-937 cells). The small samples (which correspond to bulk cell lysates diluted to single-cell level) cluster with the corresponding carrier samples, indicating that relative protein quantification from all samples is consistent and based on cell type. All quantified proteins were used for this analysis, and each protein was normalized separately for the carrier channels and the small sample channels

counting statistics while remaining comparable to that of single cells, and thus likely to be within the linear range of quantification. The reference channel for the experiments involving monocytes and macrophages is composed of an equal number of cell equivalents of monocyte and macrophage cell material (2.5 cell equivalents each) from cells counted by hemocytometer, mixed, and prepared in bulk.

### Increased throughput and improved quantification
SCoPE2 introduces a shorter nLC gradient, which allows analyzing more cells per unit of time. Furthermore, improved peptide separation and a narrower isolation window (0.7 Th) result in improved ion isolation and thus improved quantification, see Fig. 2 and Additional file 1: Fig. S5.

### Systematic parameter optimization
The sensitivity and quantification of LC-MS/MS experiments depend on numerous instrument parameters whose tuning often depends on trial-and-error approaches [25].

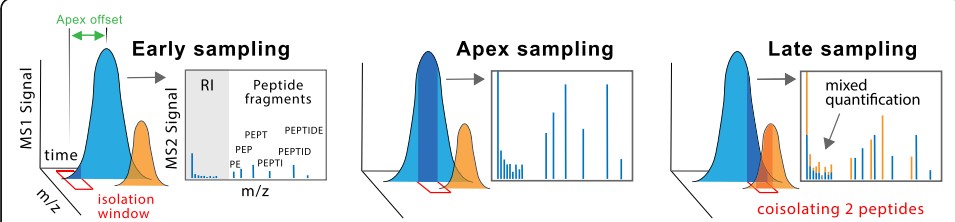

**Fig. 2** Sampling elution peaks and co-isolating precursor ions during LC-MS/MS, each labeled peptide elutes from the chromatographic column as an elution peak over a time period typically ranging from 10 to 40 s (5–20 s at mid-height) while its ions are isolated (sampled) for MS2 analysis over much shorter intervals, typically ranging from 5 to 80 ms. If the elution peak is sampled too early (left panel) or too late (right panel), the fraction of the peptide ions used for quantification and sequence identification is smaller compared to sampling the apex (middle panel). To increase the fraction of sampled ions per peptide, we used DO-MS to increase the probability of sampling the apex (Fig. 3c), decreased the elution peak width (see the "Methods" section), and increased the MS2 fill time to 300 ms. SCoPE2 quantifies peptides sequentially, one peptide at a time. For each analyzed peptide, the MS instrument aims to isolate only ions from the peptide by applying a narrow mass filter (m/z isolation window) denoted by a red rectangle in the sketch above. Yet, ions from other peptides might also fall within that window and thus become coisolated, as shown with the blue and orange peptides in the third panel. Since coisolated peptides contribute to the measured reporter ions (RI), coisolation reduces the accuracy of quantification. To minimize coisolation, we reduced the isolation windows to 0.7 Th and improved apex targeting as described above. The success of these optimizations was evaluated by the precursor ion fraction (PIF), a benchmark computed by MaxQuant as an estimate for the purity of the ions isolated for fragmentation and MS2 analysis, Fig. 3d

For example, tuned parameters may increase the number of chromatographic peaks sampled at their apexes, thus increasing the copies of peptide ions used for quantification, see Fig. 2. To make such optimizations more systematic, SCoPE2 employs DO-MS to interactively display data required for the rational optimization of instrument parameters and performance.

### Enhanced peptide sequence identification
Once the MS data are acquired, SCoPE2 can use additional features of the data to enhance their interpretation. Specifically, SCoPE2 uses a Bayesian framework (DART-ID) to incorporate retention time information for increasing the confidence of assigning peptide sequences to MS spectra while rigorously estimating the false discovery rate [26].

### Optimizing and evaluating MS analysis with standards modeling SCoPE2 sets
The quality of LC-MS/MS data strongly depends on numerous interdependent parameters (e.g., chromatographic packing, LC gradient steepness, and ion accumulation time), see Fig. 2. To optimize such parameters, we applied DO-MS on replicate samples, termed "master standards." Each 1xM standard is a 1% injection of a bulk sample (100xM) composed of eight TMT-labeled (barcoded) samples: two 5000-cell carrier samples (one for each cell type) and six 100-cell samples (three for each cell type), as shown in Fig. 1b. Each 1-µl injection of this bulk sample constitutes a 1xM standard which contains peptide input equivalent to 50 cells in each carrier channel, with the remaining six channels each containing peptide input equivalent to a single cell. Thus, the 1xM standards approximate idealized SCoPE2 standards that enabled us to focus on optimizing LC-MS/MS parameters using identical samples, i.e., independently of the biological variability between single cells.

First, we optimized our analytical column configuration and LC gradient settings. Each 1xM sample was analyzed over a 60-min active gradient since our goal was to optimize the number of proteins quantified across many cells, rather than merely the number of proteins quantified per sample [6]. By varying chromatographic parameters and benchmarking their effects with DO-MS, we minimized elution peak widths. Sharper elution peaks increase the sampled copies of each peptide per unit time and reduce the probability that multiple peptides are simultaneously isolated for MS2 analysis, see Fig. 2. Concurrent with optimizing peptide elution profiles, we optimized the data-dependent acquisition MS settings, such as minimum MS2 intensity threshold, MS2 injection time, and the number of ions sent for MS2 analysis per duty cycle (i.e., TopN), to increase the probability of sampling the apex of the elution peak of each peptide. This optimization increased the number of ion copies sampled from each peptide [25].

The 1xM standards also permitted estimating the instrument measurement noise—independently of biological and sample preparation noise—in the context of diluted bulk standards modeling SCoPE2 sets. This noise estimate was motivated by our concern that factors unique to ultra-low abundance samples, such as counting noise [1, 4, 6, 29], may undermine measurement accuracy. To estimate the measurement reproducibility, we correlated the RI intensities of lysates corresponding to the same cell type: an average correlation of 0.98 suggested good reproducibility. However, this reproducibility benchmark cannot evaluate the accuracy of relative protein quantification, i.e., the ability of SCoPE2 to quantify changes of proteins across cell types [8, 30, 31]. To benchmark accuracy, we compared the fold changes of proteins between human embryonic kidney cells and monocytes (HEK-293/U-937 protein ratios) estimated from two cell lysates diluted to single-cell levels against the corresponding ratios estimated from the bulk samples used as carriers (Fig. 1b). The high concordance of these estimates (Spearman $\rho = 0.84$) strongly indicates that the instrument noise in quantifying proteins from standards with isobaric carriers is small, consistent with our arguments that the abundance of proteins in single mammalian cells is high enough to minimize the sampling (counting) noise [6].

To further evaluate relative quantification, beyond the results for two samples diluted to single-cell level (Fig. 1b), we consolidated the data from 73 1xM standards and computed all pairwise correlations. This 584-dimensional matrix was projected onto its first two principal components (PC). The largest PC accounts for 64% of the total variance in the data and perfectly separates all samples corresponding to HEK-293 cells or monocytes. Crucially, the cell lysates diluted to single-cell levels cluster the same way as the corresponding carrier samples, indicating that the clustering is driven by cell type-specific protein differences rather than by reproducible artifacts. Thus, the standards with isobaric carriers (modeling SCoPE2 sets) can reliably quantify protein abundances at the single-cell level.

### Model system and technical benchmarks

Having demonstrated that proteins from 1xM standards can be quantified with low noise, we next applied SCoPE2 to the analysis of single cells. As a model system, we chose monocytes differentiated into macrophage-like cells in the presence of an agonist

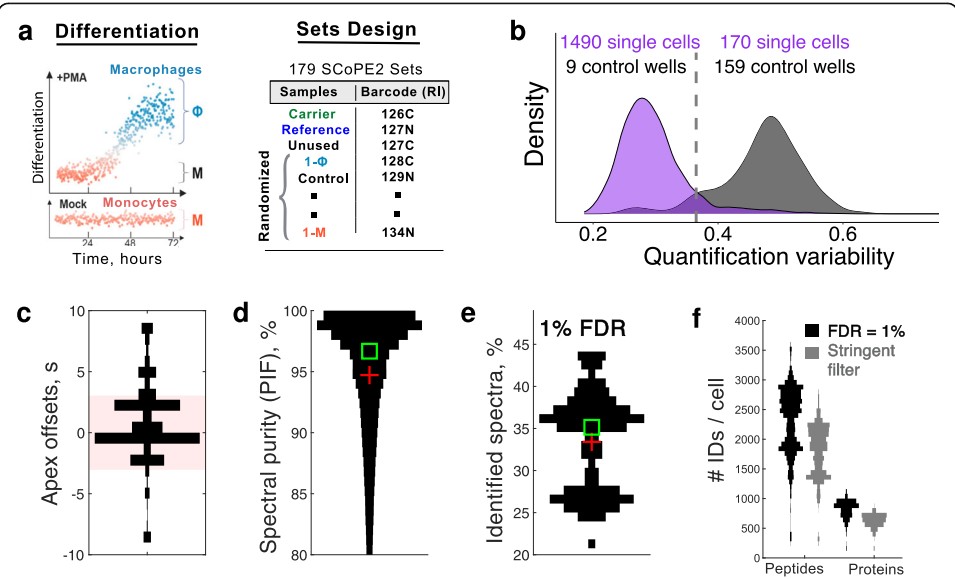

**Fig. 3** Model system and technical benchmarks for analyzed single cells and proteins. **a** Monocytes were differentiated into macrophages by PMA treatment, and FACS-sorted cells prepared into 179 SCoPE2 sets, labeled with TMT 11-plex or TMT 16-plex. **b** Distributions of coefficients of variation (CVs) for the fold changes of peptides originating from the same protein. The CVs for single cells are significantly lower than for the control wells. **c** A distribution of time differences between the apex of chromatographic peaks and the time when they were sampled for MS2 analysis, see Box 1. Over 80% of ions (shaded box) were sampled within 3 s of their apexes. **d** A distribution of precursor ion fractions (PIF) for all peptides across all SCoPE2 sets. PIF is a quantitative metric computed by MaxQuant [33] to estimate the degree of coisolation. For most MS2 scans, over 97% of the ions isolated for fragmentation and MS2 analysis belonged to a single precursor (peptide sequence), see Fig. 2. The square and the cross mark, the median, and the mean respectively. **e** About 35% of MS2 spectra are assigned to peptide sequences at 1% false discovery rate (FDR). The lower mode of the distribution corresponds to samples analyzed when the quadrupole of our instrument had suboptimal ion transmission performance. **f** The number of identified and quantified peptides and proteins in single cells from SCoPE2 sets analyzed on 60 min nLC gradients. All peptide and protein identifications are at FDR below 1% and are supported by DART-ID [26]. The criteria for stringent filtering are described in the "Methods" section. See Additional file 1: Fig. S3b for the number of peptides and proteins identified from MS spectra alone. The number of proteins with non-zero RI intensity in control wells ranged from 66 to a few hundred in contaminated or cross-labeled control wells that were excluded from the analysis. The x-axes in all violin plots (vertical histograms) correspond to counts, number of MS2 scans in **c**, number of PSMs in **d**, and number of SCoPE2 sets in **e** and **f**

of protein kinase C, phorbol-12-myristate-13-acetate (PMA), Fig. 3a. We chose this system since it provides a clear benchmark—the ability to identify two closely related but distinct cell types. This system also presents an open research question: are macrophage-like cells produced from this differentiation as homogeneous as the monocytes from which they originate or more heterogeneous? To answer this question independently from the heterogeneity inherent to primary monocytes, we used a homogeneous monocytic human cell line, U-937 [32].

The SCoPE2 workflow (Fig. 1a) can be used with manually picked cells, FACS-sorted cells, or cells isolated by microfluidic technologies that minimize the volume of the droplets containing cells [6]. Here, we used a BD FACSAria I cell sorter to sort single cells into 384-well plates, one cell per well, see Fig. 3a and Additional file 1: Fig. S1. The single cells from two biological replicate preparations of the differentiation protocol were sorted and prepared into 179 SCoPE2 sets, as depicted in Fig. 1a and Fig. 3a. The sorting followed a randomized layout to minimize biases, and sample preparation was automated as described in the "Methods" section. Each set has an isobaric carrier

corresponding to about 200 cells and a reference corresponding to 5 cells. The size of the carrier was chosen to optimize the number of quantified proteins and the number of copies sampled per protein based on guidelines from controlled experiments [29]. The carriers of some sets were sorted individually, with 200 cells per well, while the carriers and reference channels of other sets were diluted from a larger bulk sample. We analyzed two biological replicates available at massIVE [34, 35]: the first used TMT 11-plex [34], and the second used TMT pro 16-plex [35].

First, we sought to identify successfully analyzed single cells based on low variability for the relative protein quantification derived from different peptides, Fig. 3b. Specifically, different peptides originating from the same protein should provide similar estimates for the protein fold changes across single cells, as observed in SCoPE2 data (Additional file 1: Fig. S6). The variability between estimates from different peptides was quantified by the coefficient of variation (CV) for all peptides originating from the same protein, i.e., standard deviation/mean for the RI ratios. Then, the median CV (across all proteins) of a single cell provides a measure for the consistency of relative protein quantification in that cell.

To estimate the background noise, we incorporated control wells into SCoPE2 sets, Fig. 2a. These control wells did not receive cells but were treated identically to wells with single cells, i.e., received trypsin, TMT additions, and hydroxylamine. Thus, any reporter ion intensities detected in the control wells should correspond to the coisolation and background noise; this noise should be highly variable and uncorrelated for different peptides originating from the same protein because the peptides likely elute at different points in the chromatography and have different mass-to-charge ratios. This high variability is reflected in the high median CV for the control wells, higher than for single-cell wells. The distribution of CVs for all single cells and control wells indicates that 1490 single cells have lower quantification variability than the control wells, Fig. 3b. These single cells were analyzed further, see the "Evaluating single-cell quantification" section.

As explained in Fig. 2, the quantitative accuracy of LC-MS/MS measurements depends on sampling the apex of chromatographic peaks. Thus, we used DO-MS to optimize the instrument parameters so that chromatographic peaks are sampled close to their apexes [25]. As a result, most ions were sampled for MS2 analysis within 3 s of their apexes (Fig. 3c). The combination of sampling close to the apex, sharp chromatographic peaks, and narrow isolation widows (0.7 Th) minimized the simultaneous coisolation of different peptide ions for MS2 analysis. Indeed, MaxQuant estimated over 97% median purity of the ions sent for MS2 analysis, Fig. 3d. Independent estimates of coisolation by Proteome Discoverer confirm the purity of MS2 spectra, Additional file 1: Fig. S5b. The high spectral purity and the use of retention times to bolster peptide sequence identification [26] allowed assigning a peptide sequence to about 35% of the MS2 spectra from each SCoPE2 run at 1% FDR, Fig. 3e. The high spectral purity supports accurate quantification, consistent with a linear dependence between the measured signal and the input amount shown in Additional file 1: Fig. S3a.

On average, SCoPE2 quantifies over 2500 peptides corresponding to about 1000 proteins per SCoPE2 set analyzed on 1-h chromatographic gradient, Fig. 3f. While longer gradients can increase this coverage, they will reduce the number of cells analyzed per unit time. Since most single-cell analysis requires analyzing large numbers of single

cells, we focused on shorter nLC gradients that increase the number of proteins quantified across many single cells rather than merely the number of proteins per cell [6]. We applied further filtering of peptides and proteins to ensure FDR below 1% within each peptide, and within each protein, not just across all peptides and data points; see the "Methods" section. Not all proteins are quantified in all single cells because some protein values are missing. The first type of missing values is due to peptides sent for MS2 analysis but resulting in too low signal-to-noise ratio (SNR) for quantification. This is a minor contribution to missingness. Indeed, only 10% of the data for peptides identified in a SCoPE2 set were missing because the RI signal in some single cells is below the SNR threshold. This missingness can be reduced further by increasing ion sampling (accumulation) times or by narrowing chromatographic peaks, see Fig. 2 [6, 29]. The second and major type of missing values is due to peptides not sent for MS2 analysis. This is because the MS instrument does not have time to send for MS2 scans all ions detected at MS1 survey scans, and thus, it isolates for MS2 analysis different peptide subsets during different SCoPE2 runs (see Fig. 1a and Fig. 2), a well-described phenomenon for data-dependent acquisition (DDA) [11, 12]. As a consequence, we obtained quantification for a diverse set of 3042 proteins spanning a wide dynamic range, Additional file 1: Fig. S2. Thus, the average number of data points (single cells) per protein is about 305. This type of missing data, unlike the first, is a product of experimental design and not necessarily a lack of sensitivity. To increase the reproducibility of protein measurement between SCoPE2 sets, a specific list of peptides can be targeted for analysis in each set [6, 17].

### Cell type classification

High-throughput single-cell measurements are commonly used to identify and classify cell types. Thus, we sought to test the ability of SCoPE2 data to perform such classification. As a first approach, we performed principal component analysis (PCA), Fig. 4a. Using all 3042 quantified proteins, PCA separates the cells into two mostly discrete clusters along PC1, which accounts for 29% of the total variance of the data. Color-coding the cells by their labels indicates that the clusters correspond to the monocytes and the macrophages, Fig. 4a. These results are qualitatively recapitulated if the PCA is performed on the raw data without imputation and batch correction, Additional file 1: Fig. S7. To evaluate whether we can assign cell types based on the abundance of monocyte and macrophage associated proteins, we color-coded each cell by the median abundance of proteins identified to be differentially abundant from analyzing bulk samples of monocytes and macrophages, Fig. 4b. The resulting color-coding of the cells coincides with their cell types, supporting the utility of SCoPE2 data for classifying cell types.

The macrophage cluster appears more spread-out, which might reflect either the larger number of macrophage cells or increased cellular heterogeneity during the differentiation process. To distinguish, between these possibilities, we computed the pairwise correlations between monocytes and those between macrophages, Fig. 7a. The lower pairwise-pairwise correlations between macrophages suggest that indeed differentiation increases heterogeneity, Fig. 7a.

While the PCA clustering is consistent with the known cell types, it is inadequate to benchmark protein quantification. As a direct benchmark, we averaged in silico the

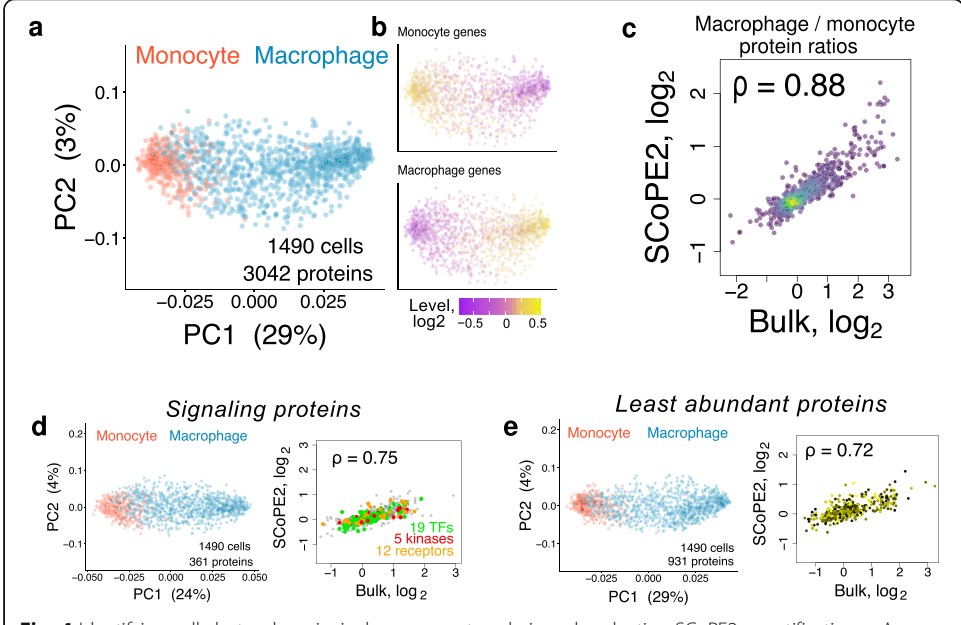

**Fig. 4** Identifying cell clusters by principal component analysis and evaluating SCoPE2 quantification. **a** A weighted principal component analysis (PCA) of 1490 single cells using all 3042 proteins quantified across multiple single cells. Missing values were imputed using $k$-nearest neighbor ($k = 3$). Cells are colored by cell type. **b** The cells from the PCA in **a** are color-coded based on the abundance of monocyte and macrophage genes, defined as the 30 most differential proteins between bulk samples of monocytes and macrophages. **c** The relative protein levels (macrophage/monocyte protein ratios) estimated from the single cells correlate to the corresponding estimates from bulk samples; $\rho$ denotes Pearson correlation. Proteins functioning in signaling (**d**) as well as the least abundant proteins quantified by SCoPE2 (**e**) allow clustering cells by cell type. The protein fold changes between monocytes and macrophages for these protein sets are consistent between single cells and bulk samples, similar to **c**

single-cell data to compute monocytes/macrophages protein fold changes and compared these estimates to the corresponding fold changes measured from bulk samples, i.e., averaging across single cells by physically mixing their lysates, Fig. 4c. The correlation ($\rho = 0.88$) indicates that SCoPE2 can accurately measure protein fold changes in single cells.

In principle, the cell type classification in Fig. 4a, b may be driven by relatively few abundant structural proteins while less abundant regulatory proteins, such as kinases, receptors, and transcription factors, might be poorly quantified. To evaluate this possibility, we applied the PCA analysis using only proteins functioning in signaling (Fig. 4d) or only the least abundant proteins quantified by SCoPE2 (Fig. 4e). The results indicate that these protein groups also correctly classify cell types (Fig. 4d, e), albeit the fraction of variance captured by PC1 is lower for the signaling proteins (24%) shown in Fig. 4d. The relative quantification of proteins from both sets correlates positively ($\rho = 0.75$, $\rho = 0.72$) to the corresponding bulk protein ratios, Fig. 4d,e.

## Macrophages exhibit a continuum of proteome states

Next, we turn to the question of whether the homogeneous monocytes differentiated to similarly homogeneous macrophages, Fig. 3a. To this end, we performed an unsupervised spectral analysis of the cell population, which allowed us to characterize cellular heterogeneity without assuming that cells fall into discrete clusters. To do so, the cells

were connected into a graph based on the correlation of their proteome profiles. We then examined the eigenvector of the graph Laplacian matrix with the smallest non-trivial eigenvalue (Fiedler vector, Eq. 1 [36]), which captures the most prominent structural axis of the cell similarity graph; see the "Methods" section. We then sorted cells according to their Fiedler vector values, thus capturing the most prominent aspect of cellular heterogeneity Fig. 5a. This cell clustering is based on hundreds of proteins with differential abundance between monocytes and macrophage-like cells, shown in Fig. 5a: Proteins with higher abundance in monocytes are enriched for proliferative functions, including chromatin organization and translation [37]. Proteins with higher abundance in macrophages are enriched for cell surface signaling and cell adhesion proteins, including intermediate filament protein vimentin, Fig. 5b. These enrichment results are consistent with the functional specialization of monocytes and macrophages and further validate the ability of SCoPE2 data to recapitulate known biology.

To explore the heterogeneity within the macrophages, we applied the same spectral analysis as in Fig. 5a, but this time only to the macrophage-like cells. The distribution

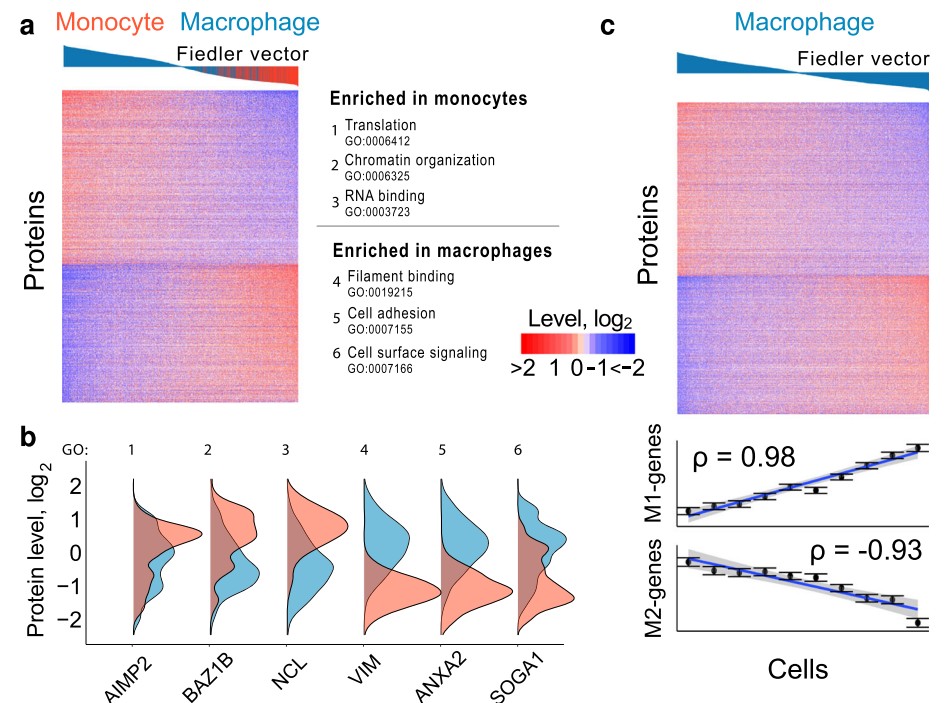

**Fig. 5** Single-cell proteomes define a continuum of macrophage polarization states. **a** Heatmap of the top 20% most variable proteins (609) between two clusters of cells identified by unsupervised spectral clustering of all quantified proteins and cells. The cells are ordered based on their rank in the corresponding Fiedler vector from the spectral clustering, see Eq. 1. The color bars above the heatmap indicate the loadings of each cell in the Fiedler vector. **b** Gene set enrichment [37] identified overrepresented functions for the proteins enriched within each cell type. These functions are displayed alongside representative protein distributions from each gene set. **c** The unsupervised spectral analysis from panel a was applied only to the macrophage-like cells, revealing a gradient of macrophage heterogeneity. Cells were ordered based on the corresponding elements of the Fiedler vector, Eq. 1. The bars above the heatmap indicate the Fiedler vector loading of each cell. The top 25% of proteins with the largest fold change between the first 40 cells and last 40 cells are displayed (761 proteins). The single-cell levels of genes in the protein data set previously reported to be enriched in M1 or M2 polarized primary human macrophages [38] are displayed at the bottom; each data point represents the median value over bins of 110 cells (1096 macrophage-like cells total), and error bars denote standard error of data points in each bin

of the elements of the associated Fiedler eigenvector (defined by Eq. 1) suggests that the cellular heterogeneity observed in this population is better described by a continuous spectrum rather than by discrete clusters, Fig. 5c. Indeed, the heatmap of protein levels for macrophage-like cells ordered based on the Fiedler eigenvector shows that most proteins change gradually across a continuous spectrum, Fig. 5c. Analyzing the proteins from this gradient, we observed a remarkable trend: genes previously identified as differentially expressed between M1 and M2-polarized primary human macrophages [38] are also differentially expressed between single macrophage cells. For example, the cells at the left edge of Fig. 5c show high expression of genes upregulated in M2-polarized macrophages, decreasing monotonically from the left to right of Fig. 5c. Genes upregulated in M1-polarized primary human macrophages appear to be expressed in a reciprocal fashion, with lower expression at the left edge of Fig. 5c, increasing monotonically across the figure. These results from Fig. 5a, c are qualitatively recapitulated if the spectral clustering is performed on the raw data without imputation and batch correction, Additional file 1: Fig. S7.

### Joint analysis of single-cell protein and RNA data

We also analyzed single cells from two biological replicates of differentiating monocytes (Fig. 3a) by scRNA-seq using the 10× Chromium platform [39] and compared the cell clustering and differential genes measured by scRNA-seq and SCoPE2. We first consider measurement noise since it may contribute to apparent differences between the protein and RNA measurements [31]. Specifically, we can expect to confidently identify differential abundance between cell types only for genes whose biological variability exceeds the measurement noise.

A major source of measurement noise for both SCoPE2 and scRNA-seq is sampling low copy number of molecules per gene. A low copy number of sampled molecules results in significant counting noise [6]. This noise arises because scRNA-seq and SCoPE2 sample only a subset of the molecules from a single cell. This sampling process contributes to a counting error: The standard deviation for sampling $n$ copies is $\sqrt{n}$ (from the Poisson distribution), and thus the relative sampling error, estimated as standard deviation over mean, is $\sqrt{n}/n = 1/\sqrt{n}$, Fig. 6a. Thus our optimization of SCoPE2 aimed to increase ion delivery not merely the number of identified peptides [25]. Similarly, to increase the sampling of mRNAs, we sequenced the 10× library to an average depth of 100,000 reads per cell. Furthermore, we chose for subsequent analysis only representative cells with at least 10,000 UMIs per cell, Additional file 1: Fig. S4.

To estimate our sampling error, we sought to convert the RI abundances (i.e., the barcodes from which SCoPE2 estimates peptide abundances) into ion copy numbers. To do so, we extracted the signal to noise ratios (S/N) for RIs and multiplied these ratios by the number of ions that induces a unit change in S/N. Since our Q-Exactive basic Orbitrap operated at 70,000 resolving power, a S/N ratio of 1 corresponds to about 6 ions [42, 43]. Thus, for our system, a S/N ratio of 50 corresponds to 300 ions; see the "Methods" section for details. The results in Fig. 6a indicate that SCoPE2 samples 10–100-fold more copies per gene than 10× Genomics, which corresponds to smaller sampling (counting) errors. The sampling can be increased further by increasing the ion sampling times, see Fig. 2.

To investigate the similarities and differences at the RNA and protein levels, we compared the correlation vectors of the scRNA-seq and SCoPE2 data as previously described [13, 40]. Specifically, for each dataset, we computed the matrices of pairwise Pearson correlations between genes, averaging across the single cells. Then to quantify the similarity of covariation of the $i$th gene, we correlated the $i$th vectors of RNA ($\mathbf{r}_i$) and protein ($\mathbf{p}_i$) correlations. The correlations between $\mathbf{r}_i$ and $\mathbf{p}_i$ for all genes are bimodally distributed, with a large positive mode, Fig. 6b. Yet, they become close to zero when the order of genes is permuted, see the null distribution in Fig. 6b. We used the two modes of the distribution of correlations between $\mathbf{r}_i$ and $\mathbf{p}_i$ to define genes with similar protein and RNA covariation (cluster 1) and genes with opposite covariation (cluster 2). The genes not included in either cluster are not differentially abundant between monocytes and macrophages at either the protein or the RNA level. The genes within a cluster correlate to each other similarly as indicated by the high correlations between $\mathbf{r}_i$ and $\mathbf{p}_i$ when these vectors are comprised only by the correlations of genes from cluster 1 or from cluster 2, Fig. 6b. To further confirm the coherence of these clusters, we displayed the RNA and protein levels of their genes as heatmaps in Fig. 6c using common PCA to order the cells [44]. Ordering the single cells and the cluster 1 genes based on the first common principal component (CPC 1) of the RNA and protein datasets confirms that the majority of genes exhibit qualitatively similar RNA and protein profiles. Gene set enrichment analysis [45] of these genes identified many biological functions, including antigen presentation, cell adhesion, cell proliferation, and protein synthesis, Additional file 1: Fig. S9. This similarity of RNA and protein profiles extends to the macrophage heterogeneity that we discovered in the SCoPE2 data Fig. 5c: The gradient is also observed with the RNA data, and it correlates to polarization marker genes albeit with smaller correlations, Additional file 1: Fig. S8. The genes from cluster 2 have opposite RNA and protein profiles and are enriched for functions including Rab GTPase activity, and protein complex assembly, Additional file 1: Fig. S10. The similarity between the RNA and protein abundances of genes from cluster 1 (Fig. 6b, c) suggests that the RNA and protein data might be projected jointly. To this end, we used Conos to generate a joint graph integrating all analyzed cells and to project them on the same set of axes, Fig. 6d–f. The joint projection results in two distinct clusters corresponding to the monocytes and macrophages, Fig. 6d. The cells analyzed by SCoPE2 are at the center of the clusters and are surrounded by more diffused clusters of the cells analyzed by scRNA-seq.

Color-coding the cells by biological and technical replicate reveals no significant clusters and thus no residual artifacts and batch effects, Fig. 6e. In contrast, color-coding the cells with monocytes and macrophage markers reveals that the clusters correspond to the cell types, Fig. 6f. These results demonstrate that, at least for our model system, the low dimensional projections of single-cell protein and RNA measurements result in similar clusters albeit the clusters based on RNA measurements are more spread out, suggesting more biological and technical variability in the RNA measurements.

## Transcriptional and post-transcriptional regulation

Next, we sought to explore transcriptional and post-transcriptional regulation based on joint RNA and protein analysis. Indeed, it has been suggested that the variability

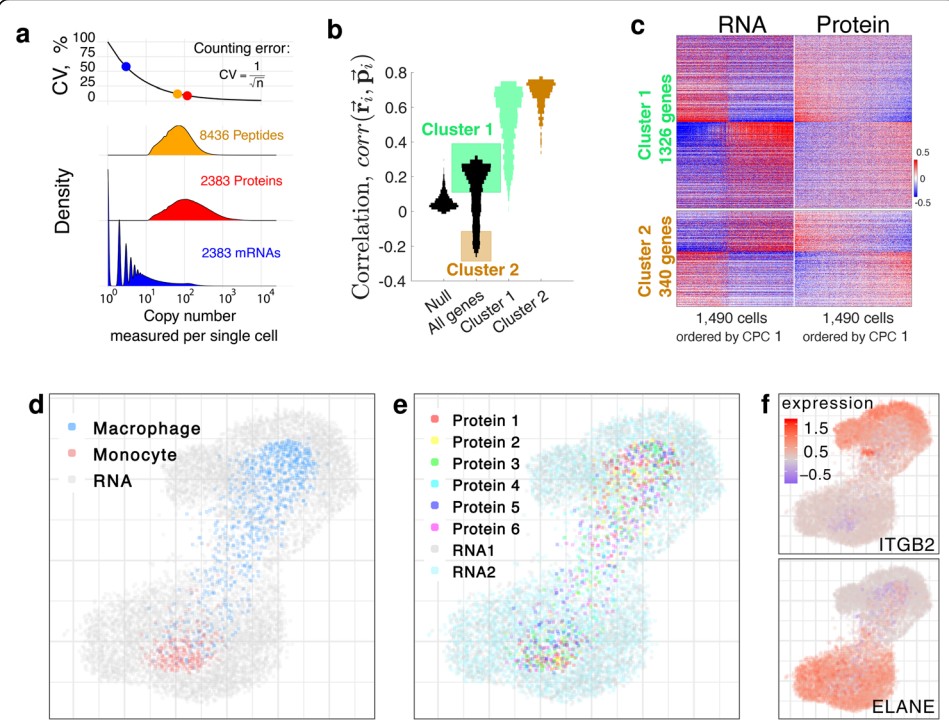

**Fig. 6** Joint analysis of single-cell RNA and protein data. **a** The number of unique barcode reads per mRNA or ions per peptide/protein for the set of 2383 genes detected in both datasets. Peptides and proteins were filtered to 1% FDR. Proteins from genes not quantified in the RNA data were omitted and vice versa. The higher copy numbers measured for proteins support more reliable counting statistics compared to mRNAs. **b** Distributions of correlations between $\mathbf{r}_i$ and $\mathbf{p}_i$, where $\mathbf{r}_i$ is the vector of pairwise correlations of the $i$th RNA to all other RNAs, and $\mathbf{p}_i$ is the vector of pairwise correlations of the $i$th protein to all other proteins [13, 40]. The null distribution corresponds to permuting the order of RNAs and proteins. The two modes of the distribution of correlations for all genes were used to define gene clusters 1 and 2. The correlations between $\mathbf{p}_i$ and $\mathbf{r}_i$ were then recomputed just within the space of genes from clusters 1 or from clusters 2 and displayed as separate distributions. **c** Genes from cluster 1 display similar abundance profiles at both the RNA and protein levels, while genes from cluster 2 display the opposite profiles. The columns correspond to single cells ordered by the first common principal component (CPC 1), which strongly correlate to cell type both for the RNA and for the protein dataset. Cluster 1 genes are ordered by the left CPC 1, and cluster 2 genes by the fold change across the ordered cells from both RNA and protein datasets. **d**–**f** Joint projections of the RNA and protein data by Conos [41]. **d** Cells analyzed by SCoPE2 are color-coded by cell type while cells analyzed by scRNA-seq are marked gray. **e** All single cells are color-coded by biological replicate and batch. **f** Cells are color-coded by the expression of marker genes for monocytes and macrophages

between single cells can be used to infer gene regulation [46, 47]. For example, transcriptional regulation should be reflected in the correlations between transcription factors (TFs) and the abundance of mRNAs whose transcription they regulate while post-transcriptional regulation can be seen in the joint distributions of RNA and protein abundances of the same gene. For such analysis, we needed to pair cells analyzed by 10× Genomics and by SCoPE2. To pair cells in similar states, we used the observations that proteins and transcripts of many genes covary together as shown in Fig. 6. Thus, we ordered the cells based on the loadings of their common first principal components and paired cells having the same rank as shown in Fig. 7a for NCL. For this gene, the protein and RNA abundances exhibit similar trends as reflected in a correlated joint distribution, Fig. 7a. This pattern is typical for abundant genes from cluster 1 as defined in Fig. 6b. Some less abundant genes from cluster 1 also exhibit similar trends though with more discrete variability at the RNA level, as exemplified with RHOC, Fig. 7b.

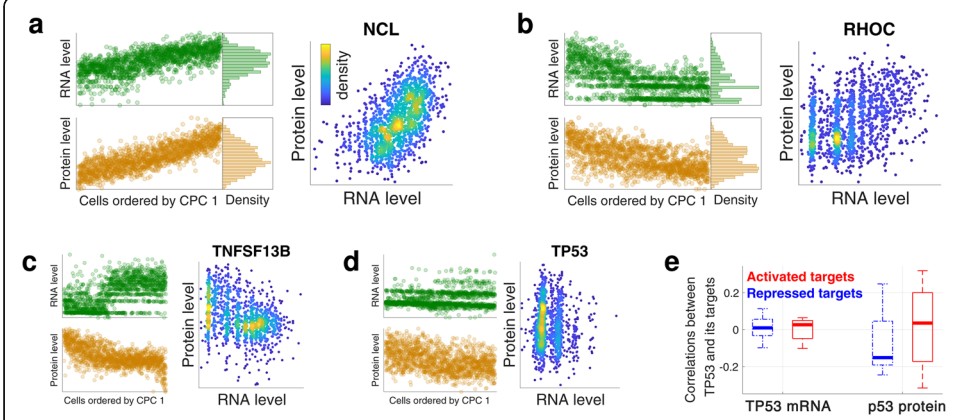

**Fig. 7** Exploring transcriptional and post-transcriptional regulation in single cells. **a** A total of 1490 cells analyzed by 10× Genomics were ordered based on the loadings of their first common principal component (CPC 1) and the abundance of NCL mRNA displayed. Similarly, 1490 cells analyzed by SCoPE2 were ordered based on CPC 1, and the abundance of NCL protein displayed. Cells having the same rank were paired to display the joint distribution of NCL. **b**–**d** The same analysis as in **a** was performed for RHOC, TNFSF13B, and TP53. **e** Distributions of correlations between p53 protein (or the TP53 RNA) and the RNA levels of its target genes. The target genes are subset into those whose transcription is repressed by p53 (blue dotted boxplots) and those whose transcription is activated by the p53 (red boxplots). The correlations for p53 protein and TP53 mRNA are significantly ($p < 0.001$) different

Other genes (particularly from cluster 2 as defined in Fig. 6b) show more divergent trends in their RNA and protein profiles as exemplified by TNFSF13B (Fig. 7c) and by the tumor suppressor protein p53 (Fig. 7d). This divergence in the levels of p53 protein and its transcript levels suggests post-transcriptional regulation, consistent with previous research demonstrating that p53 levels are regulated by protein degradation [48].

Having paired RNA and protein levels, we next explored the correlations expected from the transcriptional regulatory network. Specifically, we used the manually curated database TRRUST [49] to identify target genes whose transcription is either activated or repressed by TFs. We focused on TFs and their targets that are detected both by 10× Genomics and SCoPE2 and are known to be either activated or repressed by the TFs. We correlated the abundance of these TFs (at either mRNA or protein level) to the abundance of RNAs whose transcription they regulate. For p53 (encoded by TP53), the protein abundance correlates positively to the activated targets and negatively to the repressed targets, Fig. 7e. Other trans factors (including RELA, E2F1, and SIRT1) also correlate to their target genes as expected (negatively to repressed targets and positively to activated targets), but the number of transcripts characterized in the TRRUST database [49] and quantified in our dataset is too small to establish the statistical significance of their trends. In contrast, the correlations between the mRNA coding for p53 differ from the expectation (Fig. 7c), consistent with recent observations [50].

These results suggest that the measured protein abundances may be more informative for TF activity. Indeed, p53 is known to be regulated by proteolysis, which alters its protein levels without changing RNA abundances [48]. All correlations between the TFs and their target RNAs are weak in our dataset, which likely reflects both measurement noise and unaccounted for regulatory mechanisms, such as nuclear localization, post-translational modifications of these TFs, and additional regulators of the transcription and degradation same RNAs.

**Table 1** SCoPE2 improves quantitative accuracy, depth of proteome coverage, ease of sample preparation, and cost-effectiveness 2–10-fold over SCoPE-MS. Quantitative accuracy is evaluated by comparing relative protein ratios from in silico averaged single cells to bulk data, shown in Fig. 4c. The number of single-cell measurements per hour was estimated considering both filtering levels described in the "Methods" section. The cost of SCoPE-MS does not include the cost of sonication instrumentation. The throughput estimates for SCoPE2 are based on TMT 16-plex

| Benchmark | SCoPE-MS | SCoPE2 | Relevant figure |
|---|---|---|---|
| Correlation to benchmark fold changes | 0.2 | 0.89 | 4c |
| Purity of ions isolated for quantification | 79% | 97% | 3d |
| **Single-cell protein measurements/h** | | | |
| - "1% FDR" filtering | 610 | 5213 | 3f |
| - "Stringent" filtering | N/A | 3417 | 3f |
| **Sample preparation** | | | |
| - Time, h/cell | < 1 | < 0.03 | 3a |
| - Cost, USD/cell | < 10 | < 1 | 1a |
| **LC-MS/MS** | | | |
| - Time, h/cell | 0.50 | 0.12 | 3a |
| - Cost, USD/cell[†] | 48–96 | 10–20 | 1a |

[†]LC-MS/MS cost assumes MS facility fee of $100–200 per hour

## Discussion

SCoPE2 substantially advances the capabilities of SCoPE-MS, Table 1; it enables scalable, robust, and affordable quantification of about 1000 proteins per single cell and over 3000 proteins across many cells. This coverage is achieved with 90 min of analysis time per SCoPE2 set (about 6 min/cell), which allowed us to analyze over a thousand of cells on a single instrument in about 10 days. SCoPE2 succeeded in delivering and quantifying hundreds of ion copies from most detected proteins. This observation strongly supports the feasibility of single-cell LC-MS/MS protein quantification without amplification. Indeed, the protein fold changes estimated from the SCoPE2 data agree well with the corresponding estimates from conventional bulk methods, Fig. 1c and Fig. 4c.

The key aims of SCoPE2 are to increase throughput while reducing cost and analysis time. These aims motivated many of our choices, including the use of commercial multiwell plates (as opposed to specialized tubes as we did previously [13]), the use of 16-plex multiplexing, and the shortening of nLC gradients to 1 h. These efforts allowed SCoPE2 to reduce the cost and time for sample preparation by over 10-fold, Table 1. It also reduced the LC-MS/MS time, and thus its cost. The cost estimates in Table 1 are based on MS facility fees and can be lower for in-house LC-MS/MS analysis. The data in Fig. 7a and indicate that U-937-derived macrophages are more heterogeneous than their precursor monocytes. Upon exposure to identical environmental conditions, single macrophage cells exhibited coordinated protein level changes, Fig. 5b. In the absence of further treatment with polarizing cytokines or lipopolysaccharide to specifically induce macrophage polarization [51], the differentiated macrophage population existed in a continuum, showing reciprocal loss or gain of proteins previously identified as enriched in M1 or M2 macrophages [38], Fig. 5b. This observation suggests that polarization might be a propensity inherent to macrophages.

The reliability of data from SCoPE2 opens the potential not only to identify and classify cell sub-populations, but to go beyond such descriptive analysis: accurate protein

quantification across thousands of single cells may provide sufficient data for studying transcriptional and post-transcriptional regulation in single cells and for inferring direct causal mechanisms in biological systems [6, 8, 52] as suggested by the results in Fig. 7. This is an exciting frontier for future analysis that will demand further improvements both in the data quality and in the analysis algorithms. The analysis must incorporate estimates of measurement reliability so that experimental errors are not misinterpreted as post-transcriptional regulation [31, 52].

To have such an impact, SCoPE2 analysis must be robust and accessible. A step in this direction is replacing the expensive and time-consuming lysis used by SCoPE-MS [13] with mPOP [24], Fig. 1a. Another step is the implementation of DO-MS, which makes it easier to execute and adapt SCoPE2 to different samples and LC-MS systems [25]. A further step is the analysis identifying successful cells shown in Fig. 3b. These steps bring us closer to the transformative opportunities of single-cell proteomics [6, 8].

## Conclusion

SCoPE2 increases the accessibility, throughput, and accuracy of single-cell protein analysis by mass spectrometry. These improvements allowed us to describe heterogeneity emerging during the differentiation of myeloid cells and to explore regulatory interactions. The described methodology is general and can be broadly applied to help advance descriptive single-cell studies towards quantitative exploration of molecular mechanisms.

## Methods

### Cell culture

Human embyronic kidney cells (HEK-293 cells) were grown as adherent cultures in DMEM with high glucose (Sigma-Aldrich D5796), supplemented with 10% fetal bovine serum (FBS, Millipore Sigma F4135) and 1% penicillin-streptomycin (pen/strep, Thermo Fisher 15140122). U-937 (monocytes) cells were grown as suspension cultures in RPMI medium (HyClone 16777-145) supplemented with 10% fetal bovine serum (FBS, Millipore Sigma F4135) and 1% penicillin-streptomycin (pen/strep, Thermo Fisher 15140122). Cells were passaged when a density of $10^6$ cells/ml was reached, approximately every 2 days. Monocytes were differentiated to macrophage-like cells by first adding phorbol 12-myristate 13-acetate (PMA) to the culture medium at a final concentration of 5 nM for 24 h. Then, these newly adherent cells were washed with fresh PMA-free medium and allowed to recover in PMA-free medium for an additional 48 h before harvest. Mock-treated U-937 cells were passaged with fresh PMA-free media at 24 h and harvested along with the treated cells at 72 h. The cell lines were purchased and authenticated by ATCC.

### Harvesting cells

U-937 cells that underwent PMA-induced differentiation were washed twice with ice-cold phosphate-buffered saline (1× PBS) and dissociated by scraping. Cell suspensions of undifferentiated U-937 cells were pelleted and washed quickly with cold PBS at 4 °C media and were removed from adherent cultures of HEK-293 cells via pipetting, and

the cells were subsequently incubated with 4 °C 0.05% trypsin-EDTA (Gibco, Thermo Fisher 25300054) at 37 °C for 3 min. Cold 1× PBS was added to the dissociated HEK-293 cells, which were then pelleted via centrifugation, washed with 1× PBS, and pelleted again. The washed pellets of HEK-293 and U-937 cells were diluted in 1× PBS at 4 °C. The cell density of each sample was estimated by counting at least 150 cells on a hemocytometer. When preparing cells for SCoPE2 sets, the pellets were split into two tubes: one was diluted in 1× PBS and used for cell sorting, and the other was diluted in pure water (Optima LC/MS Grade, Fisher Scientific W6500) and used for carrier and reference channel preparation in bulk.

### Sample randomization and sorting

SCoPE2 sets were designed such that, on average, there would be 5 single macrophages, 2 single monocytes, and 1 control well per set. Control wells in this context are defined as wells that experience all sample preparation steps, except that no single cell is present in the well. SCoPE2 sets were randomized over a 384-well plate such that there would be either a maximum of 32 SCoPE2 sets (when using TMTPro 16plex) or 48 SCoPE2 sets (when using TMT 11plex) produced per plate. Single U-937 monocyte and macrophage cells were isolated and distributed into 1 µl of pure water with MassPREP peptide mixture (25 fmol/µl final concentration, Waters 186002337) in 384-well PCR plates (Thermo Fisher AB1384) using a BD FACSAria I cell sorter. In 62 of the experimental sets, the carrier channels were sorted individually, such that 100 cells of macrophage type and 100 cells of monocyte type were sorted together into a single well, serving as the carrier channel for a single SCoPE2 set. The reference channel was prepared separately by harvesting 10,000 cells of each type into a 500-µl Eppendorf tube, which was then used in all sets across multiple plates.

### Carrier and reference channel preparation in bulk

Except for the 62 experimental sets referenced above, the carrier channel was prepared in bulk and aliquoted into carriers corresponding to 200 cells each. A single-cell suspension of about 22,000 cells was transferred to a 200-µl PCR tube (USA Scientific 1402-3900) and then processed via the SCoPE2 sample preparation as described below. This cell lysate was used to generate both the carrier and reference channels.

### SCoPE2 sample preparation

Cells were lysed by freezing at − 80 °C for at least 5 min and heating to 90 °C for 10 min. Then, samples were centrifuged briefly to collect liquid; trypsin (Promega Trypsin Gold, Mass Spectrometry Grade, PRV5280) and triethylammonium bicarbonate buffer (TEAB, Millipore Sigma T7408-100ML) were added at final concentrations of 10 ng/µl and 100 mM, respectively. The samples were digested for 3 h in a thermal cycler at 37 °C (BioRad T1000). Samples were cooled to room temperature before TMT labeling (TMT11 plex kit & TMT- Pro 16plex, Thermo Fisher, Germany). Single cells and carrier cells sorted into the plate were labeled with 1 µl of 22 mM TMT label for 1 h at room temperature. For those samples in which the carrier and reference were both prepared in bulk, half of the cells were labeled with 85 mM TMT 126C (representing the carrier channel) and half were labeled with 85 mM TMT 127N (representing the

reference channel). The unreacted TMT label in each sample was quenched with 0.5 μl of 0.5% hydroxylamine (Millipore Sigma 467804-10ML) for 45 min at room temperature. Samples were centrifuged briefly following all reagent additions to collect liquid. The samples corresponding to either one TMT11plex or one TMTPro set were then mixed in a single glass HPLC insert (Thermo Fisher C4010-630) and dried down to dryness in a speed-vacuum (Eppendorf, Germany) and either frozen at − 80 °C for later analysis or immediately reconstituted in 1.2 μl of 0.1% formic acid (Thermo Fisher 85178) for mass spectrometry analysis.

### 1xM standard preparation

HEK-293 and U-937 cells were harvested and counted as described above. A total of 40,000 cells of each type were resuspended in 20 μl of HPLC-grade water (Fisher Scientific W6500), frozen at − 80 °C for 30 min, and heated to 90 °C for 10 min in a thermal cycler, before being brought to room temperature; 1 μl of benzonase (Millepore Sigma E1014-5KU), diluted to 5 units/μl, was added to each sample. The samples were then placed in a sonicating bath for 10 min. After sonication, 4 μl of a master mix containing 2.5 μl of 1 M triethylammonium bicarbonate (TEAB, Millipore Sigma T7408-100ML), 1.3 μl of 200 ng/μl trypsin (Promega Trypsin Gold, Mass Spectrometry Grade, PRV5280), and 0.2 μl of HPLC-grade water were added to each sample, and the samples were placed in a preheated 37 °C thermal cycler (BioRad T100) to digest for 3 h. 3.13 μl of each cell type was aliquoted into 4 PCR tubes, and each of the eight samples was then labeled following the rubric shown in Fig. 1b with 1.6 μl of 85 mM TMT11plex tags for 1 h at room temperature. The unreacted TMT labels in each sample were then quenched with 0.7 μl of 1% hydroxylamine solution (Millipore Sigma 467804-10ML) for 30 min at room temperature. Both labeled carrier samples were combined in a single glass HPLC insert (Thermo Fisher C4010-630), and the non-carrier samples were each brought up to 50 μl total volume, of which 1 μl of each was added to the HPLC insert. The combined samples in the HPLC insert were then dried down to dryness in a speed vacuum (Eppendorf, Germany) prior to storage at − 80. Before analysis by LC-MS/MS, the sample was reconstituted in 100 μl of 0.1% formic acid (Thermo Fisher 85178), such that a 1-μl injection containing material equivalent to 50 cells in the two carrier channels and 1 cell in each of the six additional channels was analyzed by LC-MS/MS.

### Ladder experiments preparation

U-937 cells were counted as described above, digested as described above, and serially diluted into 2000, 10, 20, 30, 40, 50, and 60-cell equivalents which were each labeled with TMT11plex tags in two randomized designs with three replicates for each design, for a total of six replicates. These samples were diluted 10× so material corresponding to 200; 1, 2, 3, 4, 5, and 6-cell equivalents were injected and analyzed by LC-MS/MS.

### SCoPE2 mass spectrometry analysis

SCoPE2 samples were separated via online nLC on a Dionex UltiMate 3000 UHPLC; 1 μl out of 1.2 μl of the sample was loaded onto a 25 cm × 75 1 μm IonOpticks Aurora Series UHPLC column (AUR2-25075C18A). Buffer A was 0.1% formic acid in water

and buffer B was 0.1% formic acid in 80% acetonitrile / 20% water. A constant flow rate of 200 nl/min was used throughout sample loading and separation. Samples were loaded onto the column for 20 min at 1% B buffer, then ramped to 5% B buffer over 2 min. The active gradient then ramped from 5% B buffer to 25% B buffer over 53 min. The gradient then ramped to 95% B buffer over 2 min and stayed at that level for 3 min. The gradient then dropped to 1% B buffer over 0.1 min and stayed at that level for 4.9 min. Loading and separating each sample took 95 min total. All samples were analyzed by a Thermo Scientific Q-Exactive mass spectrometer from minutes 20 to 95 of the LC loading and separation process. Electrospray voltage was set to 2200 V and applied at the end of the analytical column. To reduce atmospheric background ions and enhance the peptide signal-to-noise ratio, an Active Background Ion Reduction Device (ABIRD, by ESI Source Solutions, LLC, Woburn MA, USA) was used at the nanospray interface. The temperature of the ion transfer tube was 250 °C, and the S-lens RF level was set to 80. After a precursor scan from 450 to 1600 m/z at 70,000 resolving power, the top 7 most intense precursor ions with charges 2 to 4 and above the AGC min threshold of 20,000 were isolated for MS2 analysis via a 0.7 Th isolation window with a 0.3 Th offset. These ions were accumulated for at most 300 ms before being fragmented via HCD at a normalized collision energy of 33 eV (normalized to m/z 500, $z = 1$). The fragments were analyzed at 70,000 resolving power. Dynamic exclusion was used with a duration of 30 s with a mass tolerance of 10 ppm.

### Analysis of raw MS data

Raw data were searched by MaxQuant [33, 53] 1.6.2.3 against a protein sequence database including all entries from the human SwissProt database (downloaded July 30, 2018; 20,373 entries) and known contaminants such as human keratins and common lab contaminants (default MaxQuant contaminant list). MaxQuant searches were performed using the standard workflow [54]. We specified trypsin/P digestion and allowed for up to two missed cleavages for peptides having from 7 to 25 amino acids. Tandem mass tags (TMT 11plex or TMTPro 16plex) were specified as fixed modifications. Methionine oxidation (+ 15.99492 Da), asparagine deamidation (+ 0.9840155848 Da), and protein N-terminal acetylation (+ 42.01056 Da) were set as variable modifications. As alkylation was not performed, carbamidomethylation was disabled as a fixed modification. Second peptide identification was disabled. Calculate peak properties was enabled. All peptide-spectrum-matches (PSMs) and peptides found by MaxQuant were exported in the evidence.txt files. The evidence file in each massIVE repository contains the output for all RAW files in the corresponding repository. False discovery rate (FDR) calculations were performed in the R programming language environment using the best-peptide approach described in Savitski et al. [55].

### DART-ID search

Seventy-six replicate injections of the 1xM standards, 179 SCoPE2 sets, and 69 additional runs described in the MassIVE submission metadata were analyzed together by DART-ID [26]. A configuration file for the search is included in Supplementary information.

### Peptide and protein filtering

The subsequent data analysis was performed in the R (v3.5.2) programming language environment, and the code used is available at github.com/SlavovLab/SCoPE2/code.

MaxQuant was set to output PSMs at all confidence levels (PEP; posterior error probabilities) so that retention time information can be used to better discriminate between correct and incorrect PSMs [26]. The DART-ID algorithm uses all PSMs of a peptide to estimate its RT and then uses this RT estimate within a rigorous Bayesian framework to upgrade or downgrade the confidence of all PSMs. These updated confidence estimated DART-ID PEPs are then used to filter peptides and proteins to 1% FDR: the MaxQuant evidence.txt (with identification confidence updated by DART-ID) was filtered as described in Table 2. We started by removing SCoPE2 sets with less than 500 PSMs at any level of confidence. These 6 sets reflect incomplete injections or failed sample preparation. Next, peptides and proteins were filtered to 1% FDR across the whole data set, resulting in 6138 unique proteins supported by 25,015 unique peptides. The protein-level FDR was computed using the best-peptide approach described by Savitski et al. [55]. We further removed PSMs matched to reverse sequences and contaminants as defined by MaxQuant's default contaminants list. This level of filtering was employed to produce the "FDR = 1%" distributions in Fig. 3f.

To obtain a set of well-quantified protein across enough single cells to support quantitative analysis, we applied more stringent filters to the SCoPE2 data: peptides with precursor intensity fraction (PIF) below 80% were removed as well as peptides for which the mean reporter ion intensity across the single cells exceeded 10% of the values for the carrier channel. Furthermore, we removed from further analysis proteins quantified in fewer than 15 single cells, arriving at the final dataset used for quantitative analysis and displayed in Fig. 3f as "Stringent" filtering.

### Single-cell filtering

SCoPE2 single cells with suboptimal quantification were removed prior to data normalization and analysis based on objective criteria: The internal consistency of protein quantification for each single cell was evaluated by calculating the coefficient of

**Table 2** A summary of the filtration steps applied to the SCoPE2 data and the number of experiments, single cells, unique peptides, and unique proteins remaining in the dataset after each filtering step

|  | Runs | Single cells | Proteins | Peptides |
| --- | --- | --- | --- | --- |
| Number of runs | 179 | | | |
| Remove runs with less than 500 PSMs | 173 | | | |
| Filter to peptide and protein FDR < 1% | 173 | 1660 | 6138 | 25,015 |
| Remove reverse and contaminant hits | 173 | 1660 | 6083 | 24,743 |
| **"FDR = 1%" filtering** (Fig. 3f) | **173** | **1660** | **6083** | **24,743** |
| Remove peptide with MS2 purity (PIF) below 0.8 | 173 | 1660 | 5271 | 16,173 |
| Remove peptides with abnormally high intensity | 173 | 1660 | 4788 | 14,084 |
| Remove cells with median CV below 0.365 | 172 | 1490 | 4735 | 13,809 |
| Remove proteins quantified in less than 15 cells | 172 | 1490 | 3042 | 8304 |
| **"Stringent" filtering** (Fig. 3f) | **172** | **1490** | **3042** | **8304** |

variation (CV) for proteins (leading razor proteins) identified with > 5 peptides for that cell. The coefficient of variation is defined as the standard deviation divided by the mean. The CVs were computed for the relative reporter ion intensities, i.e., the RI reporter ion intensities of each peptide were divided by their mean resulting in a vector of fold changes relative to the mean. Control wells were used to determine a reasonable cutoff value for the median CV per cell below which we could have higher confidence that that channel truly contained cellular material and not just signal from noise or contamination. Control wells with low CV (9 wells) were removed from further analysis. Finally, proteins quantified in less than 15 single cells were removed, Table 2.

### Data transformations

After filtering, the peptide-level reporter ion intensities for the remaining single cells were arranged into a matrix of peptides x single cells (rows x columns). All single-cell reporter ion intensities were normalized by the reference channel intensities in their respective sets. The columns then the rows were normalized by dividing by their median and mean values, respectively (computed ignoring missing values). Peptides quantified in at least 15 cells were kept, with the average peptide being quantified in 286 single cells after this filtering. The values in the matrix were $log_2$ transformed, then the protein-level quantification was calculated by mapping each peptide to its respective (leading razor) protein and taking the median value if there was more than 1 peptide mapped to that protein. The resulting matrix has dimensions of proteins x single cells (rows x columns). The data was again normalized by subtracting the column then the row medians ($log_2$ scale). The average protein was quantified in 305 single cells.

### Imputing missing values

Missing values in the protein x single cell matrix were imputed by k-nearest neighbor imputation ($k = 3$) using Euclidean distance as a similarity measure between the cells.

### Weighted principal component analysis

From the protein x single cell matrix, all pairwise protein correlations (Pearson) were computed. Thus, for each protein, there was a computed vector of correlations with a length the same as the number of rows in the matrix (number of proteins). The dot product of this vector with itself was used to weight each protein prior to principal component analysis. The principal component analysis was performed on the correlation matrix of the weighted data.

### Single-cell RNA-seq data

A cellular mixture identical to that used for the single-cell proteomics was assessed with scRNA-seq using 10× Genomics Chromium platform and the Single Cell 3′ Library & Gel Bead Kit (v2). Two biological replicates of the cell suspension (stock concentration: 1200 cells/μl) were loaded into independent lanes of the device. An average number of about 10,000 cells/lane were recovered. Following the library preparation and sample QC by Agilent BioAnalyzer High Sensitivity chip, the two libraries were pooled together, quantified by KAPA Library Quantification kit and sequenced using the Illumina Novaseq 6000 system (Nova S1 100 flow cell) with the following run

parameters: Read1: 26 cycles, i7 index: 8 cycles, Read2: 93 cycles. Demultiplexing and count matrix estimation was carried out using the CellRanger software. Upon QC and manual inspection, the cells containing less than $10^4$ UMI barcodes were determined to be amplifying empty or background droplets and discarded. Joint alignment of proteome and scRNA-seq data was performed using the Conos package [41], using CCA space. The scRNA-seq datasets were pre-processed using the pagoda2 package to normalize for cell size and gene variance distributions. To ensure a balanced comparison, from each scRNA-seq replicate, we sampled the number of cells equivalent to the number of cells measured in the single-cell proteome replicates. To ensure that the sampled cells were representative, they were sampled around the mode of the scRNA-seq cell-size distribution. The proteome datasets were also pre-processed using pagoda2; however, no variance normalization was performed—the RI values were used directly for downstream alignment. Conos alignment of scRNA-seq and proteome datasets was performed using a neighborhood size of $k = 15$ using joint PCA space. The resulting alignment graph was visualized using a largeVis embedding (Additional file 1: Fig. S4).

### Converting signal-to-noise to ion counts

Signal-to-noise (S/N) was extracted from the raw files by matching the MS2 scan number of each identified peptide with the corresponding S/N ratios recorded in the raw files. The scripts for performing this are available at github.com/SlavovLab and were developed based on scripts by Sonnett et al. [56] available at scholar.princeton.edu/wuehr/tmtc. Ion counts were calculated by multiplying the S/N by an estimate for the number of ions that induces a unit change in S/N for our instrument. This factor was estimated as 3.5 for an orbitrap at a resolving power of 240,000. This factor scales with the square root of the ratio of the resolving power, thus for our instrument run at 70, 000 resolving power, the number of ions that induces a unit change in S/N is $3.5 \times sqrt(240,000/70,000) = 6.5$ [42, 43].

### Spectral clustering of cells

Spectral clustering was performed by first computing a matrix of positive pairwise weights, **W** between all cells. We defined the weight between two cells to be their Pearson correlation plus 1 so that all weights were positive. Then the Laplacian matrix is $\mathbf{L} = \mathbf{D} - \mathbf{W}$, where **D** is a diagonal matrix whose diagonal elements contain the sum of elements in the corresponding rows of **W**, i.e., $D_{i,i} = \text{sum}_j W_{i,j}$. Then trivially, the smallest eigenvalue of **L** is 0, and its corresponding eigenvector is the constant vector, e.g., the vector of ones. The second smallest eigenvalue corresponds to the Fiedler vector, the non-constant vector **v** that minimizes Eq. 1.

$$\mathbf{v}^T \mathbf{L} \mathbf{v} = \frac{1}{2} \sum_{i,j} w_{ij} (v_i - v_j)^2, \text{ so that } \mathbf{v}^T \mathbf{v} = 1 \tag{1}$$

Thus, computing this Fiedler eigenvector corresponds to global convex optimization that assigns similar **v** elements to cells connected by high weights [36]. The Fiedler vector was used for sorting cells in Fig. 5.

### Common principal component analysis

To jointly analyze the protein and RNA data, we performed common principal component analysis (CPCA) in the space of genes from cluster 1 using the Krzanowski method [44]. Specifically, we computed the correlation matrices of RNA correlations $\mathbf{R}_r$ and protein correlations $\mathbf{R}_p$ for all genes from cluster 1 and determined the eigenvector with the largest eigenvalue of the matrix $\mathbf{R}_r + \mathbf{R}_p$. This common principal component 1 (CPC 1) was used to order the genes in Fig. 6c. The single cells in all panels of Fig. 6 and Fig. 7 were ordered by the corresponding right principal components computed by multiplying each dataset with CPC 1. We performed a similar analysis ordering the cells and genes by the first principal component (or by the Fielder vector) of each dataset computed independently and observed qualitatively identical results. Similarly, CPCA performed in the space of all genes resulted in a very similar ordering of the cells and the genes. We displayed the ordering based on the CPCA performed in the space of the genes from cluster 1 because the RNA and protein data for these genes satisfy the key assumption of CPCA, namely their first principal components in the space of genes are strongly correlated. This approach is also conceptually simple (orders cells based on the majority of genes whose RNA and protein levels exhibit qualitatively similar trends) and is likely to generalize to other datasets.

### Gene set enrichment analysis

Gene set enrichment analysis was performed by comparing the distributions of abundances of proteins/RNAs from a functional group to that of all proteins/RNAs and computing a probability that these distributions are samples from the same master distribution [45, 57]. The effect size is summarized as the mean abundance of all proteins/RNAs from a functional group.

### Determining genes enriched in monocytes and macrophage-like cells from bulk proteomic data

For each protein profiled by bulk proteomic methods, a two-sided $t$ test was performed comparing the relative protein level between the two cell types (20 replicates per cell type). Fold change between the two cell types was calculated by taking the difference in means, and the top 60 most differential proteins (30 "upregulated" in monocytes, 30 "upregulated" in macrophage-like cells) with a $p$ value less than 0.01 from the $t$ test were taken. This list of proteins constitutes the "monocyte genes" and the "macrophage genes" displayed in Fig. 4b. Genes up and downregulated in M1 and M2 macrophage subtype were determined from MSigDB gene sets M13671 and M14515 [58].

### Evaluating single-cell quantification

While developing SCoPE2, we observed that many experimental failures manifest non-specifically with RI ratios between the carrier and the single-cell peptides that deviate from the expected ones, i.e., "compressed RI ratios." In particular, failures to sort the single cells or to extract and digest their proteins result in lower than expected RI intensities in the single-cell samples. Conversely, contamination of the single-cell samples or cross labeling peptides from the carrier with TMT labels for single cells can result in unusually high RI signal for single cells (i.e., "compressed RI ratios"). Thus, large

deviations from the expected relative RI intensities of single cells may indicate a variety of potential problems in sample preparation. We chose as a QC metric the consistency of relative protein quantification from different peptides as quantified by the coefficient of variation (CV). Crucially, the CV is computed from the fold changes between single cells relative to the mean across all samples. Thus, the CVs reflect the consistency of relative quantification which is relevant to our ability to discern biologically meaningful protein variation. CV computed from the non-normalized RI intensities (absolute scale RI intensities) are less useful for this analysis since they can be low even for very noisy data; they conflate different sources of variance and are dominated by the huge difference in the abundance of different proteins that is consistent across different human tissues [31].

## Supplementary Information

---

**Additional file 1.** contains **Fig. S1-S10**. Pdf file with all supplementary figures (Fig. S1-S10) with corresponding figure legends.

**Additional file 2.** Review history.

---

### Acknowledgements

We thank A.T. Chen, S. Semrau, A. Marneros, A. Makarov, M. Jovanovic, J. Alvarez, Z. Niziolek, Y. Katz, A. Andersen, B. Budnik, T. Hirz, and B. Karger for the assistance, discussions, and constructive comments. This work was funded by a New Innovator Award from the NIGMS from the National Institutes of Health to N.S. under Award Number DP2GM123497, an Allen Distinguished Investigator award through The Paul G. Allen Frontiers Group to N.S., an iAward from Sanofi to N.S., and through a Merck Exploratory Science Center Fellowship, Merck Sharpe & Dohme Corp. to N.S.

### Peer review information

### Review history

The review history is available as Additional file 2.

### Authors' contributions

Experimental design: H.S. and N.S.
LC-MS/MS: H.S., D.H.P., R.G.H., and A.K.
Sample preparation: H.S., E.E., A.A.P., and M.S.
Data analysis: H.S., P.K., and N.S.
Writing and editing: H.S., E. E, and N.S.
Raising funding and supervision: N.S.
The authors read and approved the final manuscript.

### Authors' information

Twitter handles: @slavov_n (Nikolai Slavov).

### Availability of data and materials

The raw MS data and the search results were deposited in MassIVE (ID: MSV000083945 [34] and MSV000084660) [35]. Reanalysis of the raw data is available in MassIVE (ID: RMSV000000326). The raw and processed 10× Genomics data were deposited in NCBI GEO GSE142392 [39] The data are also available via a dedicated website: scope2.slavovlab.net All code can be found at the SCoPE2 GitHub repository [59] and its Zendo repository [60]. This code was used to develop the scp Bioconductor package [61], which can also be used to analyze the data and to apply the analysis to other datasets.
• Facilitating LC-MS/MS evaluation: To facilitate the evaluation of our RAW LC-MS/MS data, we include detailed distribution plots generated by DO-MS [25]. These plots allow quick assessment of the nLC, ions detected at MS1 and MS2 level, apex offsets, identification rates, and other important LC-MS/MS features.
• Facilitating data reuse: To facilitate reanalysis of our single-cell protein and RNA data, we also made them available in easily reusable text formats from scope2.slavovlab.net. For the MS data, we prepared 3 files in comma-separated values (csv) format as follows:
1. Peptides-raw.csv – peptides × single cells at 1% FDR. The first 2 columns list the corresponding protein identifiers and peptide sequences and each subsequent column corresponds to a single cell. Peptide identification is based on spectra analyzed by MaxQuant [33] and is enhanced by using DART-ID [26] by incorporating retention time information.
2. Proteins-processed.csv – proteins × single cells at 1% FDR, imputed, and batch corrected.
3. Cells.csv – annotation × single cells. Each column corresponds to a single cell and the rows include relevant metadata, such as cell type if known, measurements from the isolation of the cell, and derivative quantities, i.e., rRI, CVs, and reliability.

Supplemental website can be found at scope2.slavovlab.net

**Competing interests**
The authors declare that they have no competing financial interests.
Correspondence: Correspondence and materials requests should be addressed to nslavov@alum.mit.edu

**Author details**
[1]Department of Bioengineering and Barnett Institute, Northeastern University, Boston, MA 02115, USA. [2]Present Address: Department of Biochemistry, Centre for Proteome Research, University of Liverpool, Liverpool L69 7ZB, UK. [3]Present Address: Merck Exploratory Sciences Center, Merck Sharp & Dohme Corp., 320 Bent St., Cambridge, MA 02141, USA. [4]Department of Biomedical Informatics, Harvard Medical School, Boston, MA 02115, USA.

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

## 