## [**Additional file 2.** Review history. · Genome Biology]

Review History

First round of review

Reviewer 1

Are you able to assess all statistics in the manuscript, including the appropriateness of statistical tests used? No, I do not feel adequately qualified to assess the statistics.

Comments to author:

Specht et al describe improvements to their process for single cell proteomics. Sample handling including cell lysis and protein digestion are improved and sped up. They increased the number of peptides identified to about 2500, which is an improvement of almost 3x from the previous method. A particular strength of the SCOPE approaches is the use of TMT for multiplexing which allows increased throughput. It does not rival that of scRNA-SEQ, but it significantly improves over other single cell proteomic methods. They use the improved method to survey the proteomes of >1,000 cells of monocytes and macrophages and they observe that macrophage heterogeneity is emerging in the absence of stimulation by cytokines. In this study they also included scRNA-SEQ of monocytes and macrophages, but they are not done in the same cells. While I'm not suggesting this is being asked of the authors for this study, being able to do both in the same cell would be a significant advance. And it is feasible as this process has been done in biopsy samples using Trizol which allows extracting mRNA from protein for the analysis of both, but it was not done at the single cell level (10.1681/ASN.2009060628).

This study provides an advance over the previous method from this group and it provides a significant advance in single cell proteomics especially with the addition of scRNA-SEQ. I'm not exactly sure, but this is probably the first comparison of single cells by both methods.

Issues:

- 1) It may be in the graphs (figure 2f?), but I'm wondering about the variations in peptide and protein identifications in both controls and the monocyte/macrophage samples. I'm looking for a number like 2500 +/- 100 for peptides and proteins. The reason I ask is TMT is reputed to produce fewer missing values and I expect this would be true as well in these studies, but these studies run close to the limit of noise for the reporter ions.
- 2) Have the authors calculated a copy number limit for this method, e.g. they can detect down to X copies per cell?
- 3) It's not a big issue, but cost is rarely a scientific consideration especially during proof of concept. While it is true NIH worked hard to bring down the cost of genome sequencing, it was really the efforts of businesses that pushed it down once new approaches proved to be scientifically valid. Speed = throughput and that is fine to address.
- 4) Is the heterogeneity observed in the macrophages at the single cell level observable in an experiment performed with a mass of cells (e.g. how we would normally do the proteomic experiment)? Is the only way to see this heterogeneity with single cell proteomics?

Reviewer 2

Are you able to assess all statistics in the manuscript, including the appropriateness of statistical tests used? Yes, and I have assessed the statistics in my report.

Comments to author:

Specht et al. present a m/s which represents an overhauled version of the original SCOPE-MS method, which is a single-cell proteomics method. The authors present advances of the new method and apply them to an interesting biological question. The work fits the scope of the journal.

Before being able to support acceptance for publications I have the following comments.

MAJOR QUERIES

1. While a single cell method, it is still very confusing for the reader as to how the method is actually single-cell, in particular with respect to the reference prepped at 100xM and then used at 1xM. Please clarify further. If the method does *not* use individual cells placed into the wells for preparation (but uses a 100th of 100 cells), then it should not be called single-cell. The Method description is confusing, as it goes back and forth between sorting single cells, sorting 100 cells (carrier cells) and bulk preparation. Please clarify.
2. The authors show efforts to make data available but there are some major gaps: the MASSIVE data repository only has an 'evidence' file from MQ as the output, but it is unclear if this is the file combined from all the 169 TMT runs or just one. The reader would have to re-do the entire analysis pipeline. At least proper explanation of what the evidence file is (just for one TMT run or for all 169???) and deposition of all 169 TMT output runs would be more convincing. The authors do deposit a protein file but only AFTER imputation. This makes it impossible for this reviewer to assess the data on its own. I would like to see the protein data BEFORE imputation deposited also as this is the actual data that comes out of the analysis.
3. The data is heavily normalized and contains an imputation step. Given >1,000 samples (cells), but an average occurrence of a protein in only 213 and the authors state simply that the rest is imputed. I would like a clear justification of what seems to be massive imputation (and the protein data before and after imputation), how truthful representation is still guaranteed if for a protein, on average, 800 data points are imputed. (If this is a misunderstanding, please clarify.)
4. Similarly, while the extensive normalization is necessary to extract signals, for a reader/reviewer to assess the data quality it is essential to also show the actual data. The m/s does not contain a single plot of actual data, only plots of correlations, projections etc. The supplement needs to contain: heat maps of peptide and protein data before imputation, frequency distributions of protein abundances, actual scatter plots. I would also like to see a simple PCA of the protein data, not a convoluted extraction of the projections onto the Fiedler vector. While perhaps the first eigenvectors might not represent the desired separation, it is important to show the actual properties of the data. — The same point applies to the correlations — the authors show distributions of correlations, but I would like to see original relative plots (at least a subset) in the supplement). As the data is not deposited that went into this (unless one redoes the extensive post-processing), it is impossible to reproduce the plots!
5. For example, p. 15 "While the PCA clustering is consistent with the known cell types, it is inadequate to benchmark protein quantification. As a direct benchmark, we averaged in silico the single-cell data to compute monocytes / macrophages protein fold-changes and compared these estimates to the the corresponding fold-changes measured from bulk samples, i.e., averaging across single cells by physically mixing their lysates, Fig. 3c. The correlation (= 0.89) indicates that SCoPE2 can accurately measure protein fold-changes in single cells." — It would be good to see actual fold-change distributions, rather than correlations of ratios etc. (in other words, the authors show plots only for the 5th analysis step, but not for the first...). This does not give a good impression as it hinders assessment of the actual data.
6. The distributions in plots like Figure 2c-f are confusing/ill-explained. What is the scale on the x-axis? This is entirely unclear to me and hinders assessment of the figures.
7. I fail to see what the dynamic range of the detected protein concentrations is (without actual data or plots of actual data). Please include a statement/plot.

8. The benchmarking of the method, in particular figure 1C, D is NOT done on single-cell data, but diluted bulk preparation. Please justify how this is still valid with respect to learning about the quality of the single-cell preparation. p. 9 "SCOPE2 design can reliably quantify protein abundances at the single-cell level." Is in my view an overstatement given that the benchmarking was NOT done on single cells at all! This is misleading and inflation of results.

9. p. 30, Suppl. Note 2, discussion of Signal-to-Noise Ratios. This is entirely confusing. The authors state an SNR of 10 for reporter ion intensities measured. They state that this means reliable detection of fold-changes of 10x or more. My issues: 1. The plots in the paper (if at all) show fold changes of about max. 4-fold for monocytes/macrophages, so how could the method then have detected those small fold-changes? I need to see a clarification or justification as to how given this SNR the smaller fold-changes could have been detected. 2. Generally, 10x fold-change or larger is A LOT in biology, typically fold changes are much smaller. The authors cite their own paper (ref 30) to substantiate the claim, BUT: this paper is not really about fold-changes across human tissues, as it looks at protein-to-RNA ratios. I do need to see more/better references for the claim that 10x fold change "encompasses most protein fold-changes between human tissues".

MINOR QUERIES

1. It is confusing/misleading that the abstract mentions protein protein interactions as the first thing even though the m/s is not about that at all.
2. The authors claim in the text that a protein on average occurs in 300 cells, but the Methods list an average of 213. This is therefore an overstatement in the text.
3. Figure 3C uses Greek letters that are unexplained.
4. Please cite references for GSEA and Fiedler decomposition as appropriate.
5. Please complete the comparison in abstract "SCOPE2 samples 20-fold more copies per gene" - more than what? I know it's more than transcript numbers, but some readers might not.
6. Page 2 citing SCOPE-MS — please restrict this to the actual reference and avoid over-citing your own reviews/opinion articles.
7. Another example of inflation of the position of SCOPE-MS, on p. 2 "While SCOPE-MS and its ideas have been reproduced and adopted by others [13,18-23]" reference 18 is from BEFORE the SCOPE-MS paper.
8. I am somewhat conflicted with Box1 is necessary given that it is about the application of published tools. And this is not a review but a research paper. Again, it somewhat feeds into my impression of inflating work/results.

Reviewer 3

Are you able to assess all statistics in the manuscript, including the appropriateness of statistical tests used? No, I do not feel adequately qualified to assess the statistics.

Comments to author:

Specht et al. present a single cell proteomic and transcriptomic analysis, using a modification of the SCOPE-MS method, developed by the same group. In SCOPE2, they increased the throughput, reduced analysis time and cost, and reach analysis of more than 1000 cells and 2700 proteins. The first part of the manuscript mainly addresses methodological optimization and quality control, and the second part, applies the techniques to monocyte differentiation into macrophages. Furthermore, they compared their SCOPE2 results to 10x single cell RNA-seq data to show the similarities and differences between RNA and protein changes. Overall, this manuscript adds to their previous studies in terms of their ability to tackle the challenge of single cell proteomics. Slavov is undoubtedly one of the pioneers in the field, and the developments, such as the ones presented here, are of importance to the proteomics community.

Furthermore, the comparison to scRNAseq has also major implications, as it suggests advantages of the protein analyses. While I see the value of this manuscript, I think the technological advancement over previous papers is rather incremental. However, deeper investigation of the monocyte-macrophages populations, and the protein-RNA relations would strengthen the manuscript, and add substantial power to the study.

1. One of their emphases is the continuum of the macrophage states. I don't understand the novelty, especially in an artificial cell system, with an external stimulation. As any biological process, I would expect it to be gradual, and therefore in each cell, I expect to see a slightly different state. This comment is also related to the wider spread of macrophages as compared to controls. If the main finding is the difference between M1 and M2, the authors should elaborate and analyze the coordinated/uncoordinated processes within the differentiation. The heatmaps in Figure 4 do not provide sufficient information about the proteins/complexes/functions.
2. Figure 5C shows RNA/protein changes. It seems like the RNA level differences are larger than the protein changes (stronger red/blue). If this is true, the authors should elaborate on these differences. In addition, the figures should indicate what are the two parts of each heatmap, and the color code? Was the clustering performed on the RNA or protein data?
3. The FDR calculation is not clear. First, they mention that their filtrations reduced the peptide FDR to 0.01%, which means 100 fold reduction over the standard FDR. How do they determine this reduction, and why do their filtrations have such a marked effect? Second, their protein assembly seems to be done outside MaxQuant, without proper FDR control. Multiple studies in the past showed the importance of protein FDR control to eliminate propagation of false identifications. The authors have to correct these analyses to meet the standards in the field.
4. The filtration process (p27) is not clearly explained. First the authors should better explain the basis for the filtration, and further present the extent of data filtered in each step (numbers of samples/numbers of proteins).
5. Related to the previous comment, some of the filtration steps seem to be based on the assumption that the single cells are rather similar to the reference samples. The authors should explain how will these filters work when analyzing unknown populations with potentially larger variability?
6. The authors indicate that for some of the TMT sets the carriers samples were sorted (200 cell/well), while for other sets they were diluted from bulk samples. What are the differences between these approaches? Do they encounter differences in identification rates or quantification accuracy?
7. P11 the authors indicate that "any reporter ion intensities detected in the control wells should correspond to coisolation and background noise". It is not clear why co-isolation would result in contamination in the control channel. Any co-isolated peptide should also be empty there, given that the well is an empty one.
8. Many graphs are missing indications of the axes and of the units of the axes.
9. Fig. 1d the presentation is not clear. If the grey marks are the legend, they should be placed outside the graph. In addition, the crosses and circles should be more easily distinguished. Since the data points are overlapping, their differences should be emphasized (e.g. colors/separate graphs).
10. MS instrument is not indicated. They only write Q-exactive, but not which model.
11. Figure 2e, why is there a bimodal distribution of the identification rates? Are these different batches/instruments?
12. In general, the authors should present whether they encountered any batch effects related to sample plates or TMT sets (or other).
13. The authors claim that Figure S3 shows high variance of macrophages compared to monocytes, on the protein level. The figure is not convincing, and the authors have to add the statistical analysis to claim significance.

Reviewer 1:

Below we respond point-by-point to all concerns raised by Reviewer #1. The remarks and questions of Reviewer #1 are in *italics*. Our responses are in blue.

Specht et al describe improvements to their process for single cell proteomics. Sample handling including cell lysis and protein digestion are improved and sped up. They increased the number of peptides identified to about 2500, which is an improvement of almost 3x from the previous method. A particular strength of the SCOPE approaches is the use of TMT for multiplexing which allows increased throughput. It does not rival that of scRNA-SEQ, but it significantly improves over other single cell proteomic methods. They use the improved method to survey the proteomes of >1,000 cells of monocytes and macrophages and they observe that macrophage heterogeneity is emerging in the absence of stimulation by cytokines. In this study they also included scRNA-SEQ of monocytes and macrophages, but they are not done in the same cells. While I'm not suggesting this is being asked of the authors for this study, being able to do both in the same cell would be a significant advance. And it is feasible as this process has been done in biopsy samples using Trizol which allows extracting mRNA from protein for the analysis of both, but it was not done at the single cell level (10.1681/ASN.2009060628).

This study provides an advance over the previous method from this group and it provides a significant advance in single cell proteomics especially with the addition of scRNA-SEQ. I'm not exactly sure, but this is probably the first comparison of single cells by both methods.

We thank Reviewer #1 for summarizing and appreciating the improvements of our methodology. As indicated by Reviewer #1, to our knowledge this is the first large scale study reporting single-cell RNA and protein measurements for thousands of genes from the same population of cells. We previously combined RNA and protein data from similar systems¹, but the analyzed cells came from different cultures and thus our analysis was confounded by uncontrolled variables that differed between the different systems. We agree that performing the analysis of RNA and proteins in the same cell is an important further advance that is feasible and currently under development as part of another project in our group.

Issues:

1) It may be in the graphs (figure 2f?), but I'm wondering about the variations in peptide and protein identifications in both controls and the monocyte/macrophage samples. I'm looking for a number like 2500 +/- 100 for peptides and proteins. The reason I ask is TMT is reputed to produce fewer missing values and I expect this would be true as well in these studies, but these studies run close to the limit of noise for the reporter ions.

Indeed, bulk samples analyzed by TMT labeling generally have very few missing datapoints within a set. In our experiments, about 10% of the data for peptides identified in a SCoPE2 set were

missing because the RI signal in some single cells is below the SNR threshold used for reporting quantification. This missingness can be reduced by increasing ion sampling (accumulation) times.

2) *Have the authors calculated a copy number limit for this method, e.g. they can detect down to X copies per cell?*

The range of abundances for proteins quantified by SCoPE2 is displayed in new supplemental Figure S2. The floor of protein detectability and quantification with SCoPE2 (as well as any other bottom-up MS method) depends not only on the abundance of a protein but also on its sequence, i.e., number of peptides produced upon digestion and their propensities to be well-separated by the chromatography and efficiently ionized by electrospray. The implementation of SCoPE2 in this work allowed us to quantify mostly abundant proteins present at $\geq 10^5$ copies / cell and only a few proteins present at $\geq 10^4$ copies / cell (those producing the most flyable peptides); see the distribution of abundances of the quantified proteins, shown in Figure S2. However, we are confident that the core ideas underpinning SCoPE2 can extend the sensitivity to most proteins in a mammalian cell, down to proteins present at ~ 1000 copies / cell.

3) *It's not a big issue, but cost is rarely a scientific consideration especially during proof of concept. While it is true NIH worked hard to bring down the cost of genome sequencing, it was really the efforts of businesses that pushed it down once new approaches proved to be scientifically valid. Speed = throughput and that is fine to address.*

We agree that the increased throughput is more important than the decreased cost.

4) *Is the heterogeneity observed in the macrophages at the single cell level observable in an experiment performed with a mass of cells (e.g. how we would normally do the proteomic experiment)? Is the only way to see this heterogeneity with single cell proteomics?*

Yes, we think the only way to detect this heterogeneity is via single-cell analysis. The heterogeneity is also detected via single-cell RNA-seq, as shown in Figure S8.

Reviewer 2:

We thank Reviewer #2 for a very thorough reading of our paper and numerous constructive suggestions. Below we respond point-by-point to all concerns raised by Reviewer #2 and summarize the revisions in response to suggested improvements. The remarks and questions of Reviewer #2 are in *italics*. Our responses are in blue.

Specht et al. present a m/s which represents an overhauled version of the original SCOPE-MS method, which is a single-cell proteomics method. The authors present advances of the new method and apply them to an interesting biological question. The work fits the scope of the journal.

Before being able to support acceptance for publications I have the following comments.

MAJOR QUERIES

1. *While a single cell method, it is still very confusing for the reader as to how the method is actually single-cell, in particular with respect to the reference prepped at 100xM and then used at 1xM. Please clarify further. If the method does *not* use individual cells placed into the wells for preparation (but uses a 100th of 100 cells), then it should not be called single-cell. The Method description is confusing, as it goes back and forth between sorting single cells, sorting 100 cells (carrier cells) and bulk preparation. Please clarify.*

We apologize for not explaining this better: The 100xM standards diluted to 1xM are not examples of single-cell analysis. They are bulk standards used to optimize the LC-MS/MS system and benchmark quantification on bulk cell lysates diluted to single-cell levels. However, they are not single-cell experiments and do not directly evaluate the accuracy of single-cell quantification. We revised our description to emphasize this important point.

2. *The authors show efforts to make data available but there are some major gaps: the MASSIVE data repository only has an 'evidence' file from MQ as the output, but it is unclear if this is the file combined from all the 169 TMT runs or just one. The reader would have to re-do the entire analysis pipeline. At least proper explanation of what the evidence file is (just for one TMT run or for all 169???) and deposition of all 169 TMT output runs would be more convincing. The authors do deposit a protein file but only AFTER imputation. This makes it impossible for this reviewer to assess the data on its own. I would like to see the protein data BEFORE imputation deposited also as this is the actual data that comes out of the analysis.*

The evidence file in each repository contains the search results from all RAW files in the repository, and we clarified this in the paper. We fully agree that the clarity of describing the data and analysis are absolutely essential for our work to be of value. We had described the data better at the supporting website [scope2.slavovlab.net](http://scope2.slavovlab.net) than on massIVE, and now we are following the latest guidelines² to describe the metadata on public servers as well. We are very happy to report that our analysis has been independently reproduced by Christophe Vanderaa and Laurent Gatto, and their replication can be found at [uclouvain-cbio.github.io/scp/](https://uclouvain-cbio.github.io/scp/). They presented it at the third Single-cell proteomics conference: [youtu.be/XMxZkw8yorY](https://youtu.be/XMxZkw8yorY).

3. *The data is heavily normalized and contains an imputation step. Given >1,000 samples (cells), but an average occurrence of a protein in only 213 and the authors state simply that the rest is imputed. I would like a clear justification of what seems to be massive imputation (and the protein data before and after imputation), how truthful representation is still guaranteed if for a protein, on average, 800 data points are imputed. (If this is a misunderstanding, please clarify.)*

Indeed, most datapoints are imputed as is commonly done with single-cell RNA-seq analysis³⁻⁵. Of course this being an established practice for single-cell omics analysis does not make it safe. Imputation can introduce artifacts as discussed for scRNA-seq⁶. To validate that our results are not merely an artifact of imputation, we also analyzed the data without imputation. Specifically, the protein fold changes in Figure 3c and Figure 4b are estimated without any imputation. Similarly, we repeated our analysis on the raw data, without any imputation and batch correction. The results from this analysis (shown in Figure S7) qualitatively recapitulate the results from the imputed data shown in main Figures 3 and 4.

4. *Similarly, while the extensive normalization is necessary to extract signals, for a reader/reviewer to assess the data quality it is essential to also show the actual data. The m/s does not contain a single plot of actual data, only plots of correlations, projections etc. The supplement needs to contain: heat maps of peptide and protein data before imputation, frequency distributions of protein abundances, actual scatter plots. I would also like to see a simple PCA of the protein data, not a convoluted extraction of the projections onto the Fiedler vector. While perhaps the first eigenvectors might not represent the desired separation, it is important to show the actual properties of the data. — The same point applies to the correlations — the authors show distributions of correlations, but I would like to see original correlative plots (at least a subset) in the supplement). As the data is not deposited that went into this (unless one redoes the extensive post-processing), it is impossible to reproduce the plots!*

The new supplemental Figure S7a shows a simple PCA without any imputation. The cell types are separated along the first principal component albeit the fraction of the variance explained by the first principal component is lower than in the batch corrected data shown in main Figure 3a. Figure S7c,d show heatmaps of the relative protein levels without imputation and batch correlation, and these recapitulate the main trends shown in the main figures. Main figures 3c,d,e show protein ratios between macrophages and monocytes computed from the raw data (without imputation and batch correction), however these were not marked so clearly in our original submission as we used Φ to denote macrophages without explaining it. Now we improved the annotation of Figure 3c. Likewise, the distributions of protein levels for individual proteins in Figure 4b are computed from raw data (without imputation and batch correction). Raw reporter ion intensities for individual peptides from the CAPG protein are shown in supplemental Figure S6.

5. *For example, p. 15 "While the PCA clustering is consistent with the known cell types, it is*

inadequate to benchmark protein quantification. As a direct benchmark, we averaged *in silico* the single-cell data to compute monocytes / macrophages protein fold-changes and compared these estimates to the the corresponding fold-changes measured from bulk samples, i.e., averaging across single cells by physically mixing their lysates, Fig. 3c. The correlation (= 0:89) indicates that SCoPE2 can accurately measure protein fold-changes in single cells." — It would be good to see actual fold-change distributions, rather than correlations of ratios etc. (in other words, the authors show plots only for the 5th analysis step, but not for the first. . .). This does not give a good impression as it hinders assessment of the actual data.

The scatter plots in Figure 3 show the actual protein fold changes, but we did not this explain this clearly. Now in the revised manuscript we have improved the explanations. To further display more directly protein fold changes, we made a volcano plot, shown in Fig. 1, and included gene ontology enrichment showing that indeed the differences observed are consistent with the known biological differences between monocytes and macrophages. Gene ontology enrichment was performed with the online tool GOrilla⁷, and complete results are included as excel tables in GOrilla_Results.zip, and available at scope2.slavovlab.net/docs/data. Hundreds of GO terms show strong enrichment in either monocytes and macrophages. Gene ontologies enriched in macrophages include actin filament binding (GO:0051015, adjusted p-value 8e-10) and signaling receptor binding (GO:0005102, adjusted p-value 0.002), consistent with expectations⁸. Gene ontologies enriched in monocytes include nucleosome organization (GO:0034728, adjusted p-value 0.001) and protein-DNA complex assembly (GO:0071824, adjusted p-value 0.0002), consistent with dividing cells (unlike macrophages).

Figure 1. A volcano plot of macrophage vs monocyte protein levels. A multiple sample t-test was performed on the data set containing 1018 single macrophage-like cells and monocyte cells. Protein P40121 (CAPG) is indicated in green. Red line indicates a 1% FDR cutoff, with about 30 % of proteins falling above the plotted red line.

6. The distributions in plots like Figure 2c-f are confusing/ill-explained. What is the scale on the x-axis? This is entirely unclear to me and hinders assessment of the figures.

This distributions are shown as non smoothed violin plots, basically vertical histograms. The x axis corresponds to counts, number of MS2 scans in panel c, number of PSMs in panel d, and number of SCoPE2 sets in panels e and f. These clarifications were added to the figure caption.

7. *I fail to see what the dynamic range of the detected protein concentrations is (without actual data or plots of actual data). Please include a statement/plot.*

The range of abundances for proteins quantified by SCoPE2 is displayed in new supplemental Figure S2. The floor of protein detectability and quantification with SCoPE2 (as well as any other bottom-up MS method) depends not only on the abundance of a protein but also on its sequence, i.e., number of peptides produced upon digestion and their propensities to be well separated by the chromatography and efficiently ionized by electrospray. The implementation of SCoPE2 in this work, allowed us to quantify mostly abundant proteins present at $\geq 10^5$ copies / cell and only a few proteins present at $\geq 10^4$ copies / cell (those producing the most flyable peptides); see the distribution of abundances of the quantified proteins, shown in Figure S2. However, we are confident that the core ideas underpinning SCoPE2 can extend the sensitivity to most proteins in a mammalian cell, down to proteins present at ~ 1000 copies / cell.

8. *The benchmarking of the method, in particular figure 1C, D is NOT done on single-cell data, but diluted bulk preparation. Please justify how this is still valid with respect to learning about the quality of the single-cell preparation. p. 9 "SCoPE2 design can reliably quantify protein abundances at the single-cell level." Is in my view an overstatement given that the benchmarking was NOT done on single cells at all! This is misleading and inflation of results.*

The data in figure 1 are relevant only to the ability of mass-spec analysis to quantify peptides present at single-cell level, but these data do not capture errors introduced from the sample preparation of individual cells and thus do not benchmark protein quantification from single cells, just mass-spec measurement of peptides from mammalian proteomes diluted to a single-cell level. We revised the text to emphasize this. Our most direct benchmarks for quantification of relative protein levels in single cells are the scatter plots shown in Figure 3. These compare directly protein fold changes estimated from the single cells and from conventional bulk LC-MS/MS analysis.

9. *p. 30, Suppl. Note 2, discussion of Signal-to-Noise Ratios. This is entirely confusing. The authors state an SNR of 10 for reporter ion intensities measured. They state that this means reliable detection of fold-changes of 10x or more. My issues: 1. The plots in the paper (if at all) show fold changes of about max. 4-fold for monocytes/macrophages, so how could the method then have detected those small fold-changes? I need to see a clarification or justification as to how given this SNR the smaller fold-changes could have been detected. 2. Generally, 10x fold-change or larger is A LOT in biology, typically fold changes are much smaller. The authors cite their own paper (ref 30) to substantiate the claim, BUT: this paper is not really about fold-changes across human tissues, as it looks at protein-to-RNA ratios. I do need to see more/better references for the claim*

that 10x fold change "encompasses most protein fold-changes between human tissues".

We apologize for the confusion, which arises entirely from our poor wording and communication rather than from any substantive disagreement. We agree completely that “Generally, 10x fold-change or larger is A LOT in biology, typically fold changes are much smaller. ” As Reviewer #2 pointed out, our single-cell data do not have such large protein fold changes. The point of the note was to describe a limitation of our analysis: Because the reporter ions from single cells for many peptides have SNR of about 10 or even lower, the data cannot detect relative fold changes exceeding 10 fold.

MINOR QUERIES

1. It is confusing/misleading that the abstract mentions protein protein interactions as the first thing even though the m/s is not about that at all.

We removed the word “interactions” from the abstract.

2. The authors claim in the text that a protein on average occurs in 300 cells, but the Methods list an average of 213. This is therefore an overstatement in the text.

We apologize for that. We corrected the numbers in the main text and the Methods text to be the same.

3. Figure 3C uses Greek letters that are unexplained.

We explained that ρ denotes Pearson correlation and replaced Φ with macrophages.

4. Please cite references for GSEA and Fiedler decomposition as appropriate.

We provided references for the Fiedler vector⁹ and for GSEA¹⁰.

5. Please complete the comparison in abstract "SCoPE2 samples 20-fold more copies per gene" - more than what? I know it's more than transcript numbers, but some readers might not.

We completed the comparison.

6. Page 2 citing SCOPE-MS — please restrict this to the actual reference and avoid over-citing your own reviews/opinion articles.

We removed unnecessary references to our own review and opinion articles.

7. Another example of inflation of the position of SCOPE-MS, on p. 2 "While SCoPE-MS and its ideas have been reproduced and adopted by others [13,18-23]" reference 18 is from BEFORE the SCOPE-MS paper.

Reference 18 was published online by Analytical Chemistry on September 11, 2019. The SCoPE-MS paper was published by Genome Biology about 11 months earlier, on October 22, 2018. The SCoPE-MS preprint was first posted on bioRxiv on January 27, 2017.

8. I am somewhat conflicted with Box1 is necessary given that it is about the application of published tools. And this is not a review but a research paper. Again, it somewhat feeds into my impression of inflating work/results.

We agree that Box1 is reviewing concepts that mass-spectrometry experts are familiar with. We included this box because our work has attracted the interest of many biologists who cannot understand the improvements of SCoPE2 without having a short introduction to these concepts. The Box seems very helpful for them and thus increases the accessibility of our paper. We agree that the Box would be unnecessary if our paper is intended only for an audience of MS experts.

Reviewer 3:

Below we respond point-by-point to all concerns raised by Reviewer #3. The remarks and questions of Reviewer #3 are in *italics*. Our responses are in blue.

Specht et al. present a single cell proteomic and transcriptomic analysis, using a modification of the SCOPE-MS method, developed by the same group. In SCOPE2, they increased the throughput, reduced analysis time and cost, and reach analysis of more than 1000 cells and 2700 proteins. The first part of the manuscript mainly addresses methodological optimization and quality control, and the second part, applies the techniques to monocyte differentiation into macrophages. Furthermore, they compared their SCOPE2 results to 10x single cell RNA-seq data to show the similarities and differences between RNA and protein changes. Overall, this manuscript adds to their previous studies in terms of their ability to tackle the challenge of single cell proteomics. Slavov is undoubtedly one of the pioneers in the field, and the developments, such as the ones presented here, are of importance to the proteomics community. Furthermore, the comparison to scRNAseq has also major implications, as it suggests advantages of the protein analyses. While I see the value of this manuscript, I think the technological advancement over previous papers is rather incremental. However, deeper investigation of the monocyte-macrophages populations, and the protein-RNA relations would strengthen the manuscript, and add substantial power to the study.

We thank Reviewer #3 for this insightful summary and for the suggestion to expand and deepen the analysis of the joint RNA and protein dataset. The results from this additional analysis are presented in a new Figure 6, which we think substantially strengthened the paper. Additionally, when we tallied the effects of each step in the filtration process in response to Reviewer #3, we discovered that an incorrect variable type assignment by one of the R functions had resulted in a failure to import the data from about 400 single cells. We fixed this problem and repeated all analysis with these additional single cells included. All results are qualitatively unchanged. We thank Reviewer #3 for their question that helped us discover this omission in data import.

1. One of their emphases is the continuum of the macrophage states. I don't understand the novelty, especially in an artificial cell system, with an external stimulation. As any biological process, I would expect it to be gradual, and therefore in each cell, I expect to see a slightly different state. This comment is also related to the wider spread of macrophages as compared to controls. If the main finding is the difference between M1 and M2, the authors should elaborate and analyze the coordinated/uncoordinated processes within the differentiation. The heatmaps in Figure 4 do not provide sufficient information about the proteins/complexes/functions.

Yes, we used an *in vitro* system that only models *in vivo* systems and has all caveats of *in vitro* systems. It is a classical system used in thousands of experiments as an example for non-polarized macrophages, classically described as M0. To our knowledge, all previous uses of the system treated the macrophage-like cells as homogeneous and unpolarized. So, the demonstration that the

macrophage-like cells are heterogeneous and the heterogeneity is correlated to the M1-M2 axis is new to the best of our knowledge.

2. *Figure 5C shows RNA/protein changes. It seems like the RNA level differences are larger than the protein changes (stronger red/blue). If this is true, the authors should elaborate on these differences. In addition, the figures should indicate what are the two parts of each heatmap, and the color code? Was the clustering performed on the RNA or protein data?*

Indeed, the variability and the dynamic range of the fold changes of the RNA data are larger, and we commented on this in the text. We also improved the annotation of Figure 5c and clarified that the genes and cells were ordered based on the common principal components computed from both datasets. Thank you for these constructive suggestions.

3. *The FDR calculation is not clear. First, they mention that their filtrations reduced the peptide FDR to 0.01%, which means 100 fold reduction over the standard FDR. How do they determine this reduction, and why do their filtrations have such a marked effect? Second, their protein assembly seems to be done outside MaxQuant, without proper FDR control. Multiple studies in the past showed the importance of protein FDR control to eliminate propagation of false identifications. The authors have to correct these analyses to meet the standards in the field.*

Indeed, we exported data from Maxquant before the protein level FDR control. MaxQuant was set to output PSMs at all confidence levels (PEP; posterior error probabilities) so that retention time information can be used to better discriminate between correct and incorrect PSMs¹¹. The DART-ID algorithm uses all PSMs of a peptide to estimate its RT and then uses this RT estimate within a rigorous Bayesian framework to upgrade or downgrade the confidence of all PSMs. These updated confidence estimated DART-ID PEPs are then used to filter peptides and proteins to 1 % FDR. Specifically, the PEPs updated by DART-ID we used to filter for peptides with a global FDR of 1% and then applied to estimated and filter to 1% protein level FDR using the best-peptide approach described by Savitski *et al.*¹². We emphasized this in the updated methods section and included Table 2 to describe the effects of each filtering step. The code used to calculate FDR and apply filtering is on the Github repository. The reference to FDR or 0.01% was a typo: we had meant an FDR of 0.01, which is an FDR of 1%. Thank you for raising this question.

4. *The filtration process (p27) is not clearly explained. First the authors should better explain the basis for the filtration, and further present the extent of data filtered in each step (numbers of samples/numbers of proteins).*

We have clarified each step of the filtration process in the Methods and added Table 2 to present the extent of data filtered in each step.

5. *Related to the previous comment, some of the filtration steps seem to be based on the assumption that the single cells are rather similar to the reference samples. The authors should explain how*

will these filters work when analyzing unknown populations with potentially larger variability?

Indeed, we removed peptides with the mean RI intensities in single cells exceeding 10% of the carrier channel. If the single cells and the carrier are drawn from the same population, we have the expectation that the mean protein abundances in the single cells should correlate strongly with the protein abundances in the carrier. The Reviewer is very correct to point out this expectation does not hold for individual cells, which is why we sought to filter based on the mean protein level across multiple single cells, not the level in an individual cell. We had not explained this well in the Methods section, so we have added a more complete description. This filtration step removed less than 20% of the peptides and it did not alter the results qualitatively. This is the only step in the filtration process that compares single cells to the carrier. Colleagues who independently reproduced our analysis built a Bioconductor package available at uclouvain-cbio.github.io/scp/ precisely with the intention to facilitate exploration of the turning on and off each step with the analysis as they described in this talk: youtu.be/XMxZkw8yorY. So we expect (and recommend) that future applications of SCoPE2 test the robustness of their results to this (and all other) filtration steps in their analysis.

6. The authors indicate that for some of the TMT sets the carriers samples were sorted (200 cell/well), while for other sets they were diluted from bulk samples. What are the differences between these approaches? Do they encounter differences in identification rates or quantification accuracy?

We wanted to test both approaches, and they both worked well, albeit with some small trade-offs. The carriers diluted from bulk lysate provided slightly higher consistency of the peptides identified in different TMT sets, and thus fewer missing datapoints. This approach is simpler to implement. On the other hand, making the carriers from sorted cells provides built in controls for the consistency of sample preparation, some aspects of which are more readily diagnosed on the carrier samples.

7. P11 the authors indicate that "any reporter ion intensities detected in the control wells should correspond to coisolation and background noise". It is not clear why co-isolation would result in contamination in the control channel. Any co-isolated peptide should also be empty there, given that the well is an empty one.

That is an important question. The control wells contain all buffers, enzymes (e.g., trypsin) and chemical used for the single cells, and each component can contribute background peptides (such as peptides from keratin proteins, trypsin autolysis, or other protein/peptide contaminants) that will be labeled with the isobaric mass tag corresponding to the control well, and thus may contribute coisolation background.

8. Many graphs are missing indications of the axes and of the units of the axes.

Thank you for promoting us to reexamine our graphs. We added any omissions that we found.

9. *Fig. 1d the presentation is not clear. If the grey marks are the legend, they should be placed outside the graph. In addition, the crosses and circles should be more easily distinguished. Since the data points are overlapping, their differences should be emphasized (e.g. colors/separate graphs).*

Indeed, the datapoints corresponding to isobaric carriers and to cell lysates diluted to single-cell level were indistinguishable in the graph. We followed the suggestion of Reviewer #3 and used different colors, which made them distinctive. Thank you for this suggestion that significantly improved the graph.

10. *MS instrument is not indicated. They only write Q-exactive, but not which model.*

We used Q-exactive basic and clarified this in the main text and the methods section.

11. *Figure 2e, why is there a bimodal distribution of the identification rates? Are these different batches/instruments?*

The lower mode of the distribution corresponds to samples analyzed when the quadrupole of our instrument had suboptimal ion transmission performance. We kept the data because we want to convey a realistic representation of what colleagues using our methodology should expect when analyzing many hundreds of single cells rather than show only our best data.

12. *In general, the authors should present whether they encountered any batch effects related to sample plates or TMT sets (or other).*

Yes, we did observe batch effects associated with the multiwell plates and much smaller (though detectable) batch effects associated with TMT sets. These batch effects were small enough so that even without correcting them with specialized batch correction software, the first principal component of the data is dominated by the cell-type specific protein differences as shown in new supplemental figure S7.

13. *The authors claim that Figure S3 shows high variance of macrophages compared to monocytes, on the protein level. The figure is not convincing, and the authors have to add the statistical analysis to claim significance.*

We used rank sum test to compute the probability that the monocyte and macrophage distributions statistically the same (sampled from the same master distribution). All distributions, including the in-data-type comparisons (i.e. macrophage protein to monocyte protein) and across-data-type (i.e. macrophage protein to macrophage RNA), are significantly different, $p < 2.2 \times 10^{-16}$ from Wilcoxon rank sum test. The very high significance is due to the fact that each distribution has tens of thousands of correlations (all pairwise correlations between hundreds of single cells).

References

1. Budnik, B., Levy, E., Harmange, G. & Slavov, N. SCoPE-MS: mass-spectrometry of single mammalian cells quantifies proteome heterogeneity during cell differentiation. *Genome Biology* **19**, 161 (2018).
2. Perez-Riverol, Y. Towards a sample metadata standard in public proteomics repositories. en. *J. Proteome Res.* (Aug. 2020).
3. Li, W. V. & Li, J. J. An accurate and robust imputation method scImpute for single-cell RNA-seq data. *Nature communications* **9**, 1–9 (2018).
4. Van Dijk, D. *et al.* Recovering gene interactions from single-cell data using data diffusion. *Cell* **174**, 716–729 (2018).
5. Klein, A. M. *et al.* Droplet barcoding for single-cell transcriptomics applied to embryonic stem cells. *Cell* **161**, 1187–1201 (2015).
6. Andrews, T. S. & Hemberg, M. False signals induced by single-cell imputation. *F1000Research* **7** (2018).
7. Eden, E., Navon, R., Steinfeld, I., Lipson, D. & Yakhini, Z. GOrilla: a tool for discovery and visualization of enriched GO terms in ranked gene lists. *BMC Bioinformatics* **10**, 48 (2009).
8. Martinez, F. O., Gordon, S., Locati, M. & Mantovani, A. Transcriptional profiling of the human monocyte-to-macrophage differentiation and polarization: new molecules and patterns of gene expression. eng. *Journal of Immunology (Baltimore, Md.: 1950)* **177**, 7303–7311. ISSN: 0022-1767 (Nov. 2006).
9. Spielman, D. in *Combinatorial scientific computing* 18 (Citeseer, 2012).
10. Subramanian, A. *et al.* Gene set enrichment analysis: a knowledge-based approach for interpreting genome-wide expression profiles. *Proceedings of the National Academy of Sciences* **102**, 15545–15550 (2005).
11. Chen, A., Franks, A. & Slavov, N. DART-ID increases single-cell proteome coverage. *PLoS Comput Biol.* doi:[10.1371/journal.pcbi.1007082](https://doi.org/10.1371/journal.pcbi.1007082) (2019).
12. Savitski, M. M., Wilhelm, M., Hahne, H., Kuster, B. & Bantscheff, M. A Scalable Approach for Protein False Discovery Rate Estimation in Large Proteomic Data Sets. *Molecular & Cellular Proteomics* **14**, 2394–2404. ISSN: 1535-9476 (2015).

Second round of review

Reviewer 1

The authors addressed my concerns and I recommend publication.

Reviewer 3

The authors answered most of my comments. However I agree with the other reviewers that the authors should be more clear about the numbers, and limit their own self-promotion. Specifically, it is important to know the number of proteins/peptides identified per cell, not only per TMT set, and also indicate the number of proteins in the empty channels (and intensities). These are minor technical edits that do not require further evaluation on my side.